# Lunar base agent-based modeling - A benchmark for simulating crewed space missions

Raymond Vera[ID]*, Anamaria Berea‡, William G. Kennedy[ID]‡

Department of Computational and Data Sciences, College of Science, George Mason University, Fairfax, Virginia, United States of America

‡ These authors are joint senior authors on this work.
* rvera2@gmu.edu

## Abstract

Space exploration has progressed significantly since the mid-20th century, and recent technological advancements, along with the emergence of commercial space travel, have led to substantial leaps in planning for future space missions. The largest planned upcoming mission is the Artemis program, supported by NASA and the international Artemis Accords, which aims to create the first permanent human presence on the Moon and in deep space (the Moon to Mars architecture). Although human psychology and team science have been crucial for the success of past space missions, from the Apollo program and Skylab to the Space Shuttle (STS) and the International Space Station (ISS), human factors and social behavior will become even more ubiquitous and essential for space missions in the new era of commercial space. By simulating upcoming permanent space missions in an agent-based model (ABM), we can draw insights into the long-term effects of human factors and interactions in space. Drawing from the literature on proxy environments (extreme environments on Earth (i.e., Antarctica), space analogs, and past space missions), and on theories of small group complex systems and team science, we created a highly probable representation or simulation of expected social interactions between astronauts, and astronauts with the lunar environment for the Artemis program (i.e., Artemis IV (Lunar Gateway) and Artemis V (Lunar South Pole Base)). Our Lunar Base ABM explores the exogenous and endogenous factors that are more likely to lead to sustainable versus catastrophic scenarios on the Moon in the next couple of decades. The model represents astronauts using a new Agent_Astronaut framework with cognitive skills, emotional states, and personality traits to capture how social and environmental factors interact to affect mission outcomes. Monte Carlo simulations consisting of tens of thousands of iterations show trade-offs in productivity and psychological well-being. This approach demonstrates how agent-based modeling can help mission planners evaluate operational resilience, team structures, and workload dynamics in support of future lunar exploration.

**Data availability statement:** All relevant data are within the manuscript. The Lunar Base ABM code is available in the GitHub repository: https://github.com/rvera-gmu/Lunar-Base-ABM.

**Funding:** GMU ORIEI Award no. 102264.

**Competing interests:** The authors have declared that no competing interests exist.

## Introduction

From its early crewed spaceflights in the 1960s, beginning with the Mercury-Atlas 6 mission (1962) and culminating in the first crewed flight beyond Earth orbit during Apollo 8 (1968), NASA advanced the capabilities needed for human space travel, space exploration, and eventual lunar landings through the Apollo program that concluded in 1972 [1]. Humanity further expanded its footprint in space with the development of the International Space Station (ISS) initiated in the 1990s, which today represents collaboration among five partner agencies: NASA, the Canadian Space Agency, the European Space Agency, the Japan Aerospace Exploration Agency, and the State Space Corporation ("Roscosmos") of Russia. The ISS has been serving as a long-term, crewed microgravity laboratory, enabling decades of scientific research and technology demonstrations with a planned deorbit transition in the 2030s [2].

The next phase in space settlement and exploration begins with NASA's Artemis campaign, which involves a multinational effort to establish a long-term human presence on the Moon and prepares for future crewed missions to Mars. Grounded in the Artemis Accords established in 2020 [3] and in compliance with the Outer Space Treaty of 1967 [4], countries and corporations worldwide agree to a set of common principles for the governance of civil exploration and the use of outer space for the benefit of all humankind. The Artemis program includes a series of missions, which started with the uncrewed test flight of the Space Launch System (SLS) rocket and the Orion spacecraft around the Moon in 2022 (Artemis I). Artemis II is scheduled to launch in 2026 and will involve a four-person crewed test flight to 8,889 kilometers (km) beyond the Moon – the farthest humans have ever traveled in space. Afterward, Artemis III is expected to result in the first lunar landing by a human since Apollo 17 in 1972, and the first one at the lunar south pole [5].

While engineering and technology innovation is necessary for space missions, understanding human and operational dynamics is also crucial for mission success [6]. This is particularly important if the objective is to establish a long-term presence on the Moon. Human exploration and operations on the Moon can be viewed from a complex systems perspective. It entails heterogeneous agents making decisions and exhibiting behavior while interacting with one another and their environment [7]. The lunar environment also encompasses endogenous and exogenous factors that give rise to nonlinear interactions, reinforced by positive and negative feedback loops, and to unpredictable and emergent phenomena. As an effective tool for providing insight into complex systems, agent-based models (ABMs) [8] can enhance planning for future lunar exploration by simulating human factors and interactions in the Artemis mission. This paper aims to show a Lunar Base ABM framework grounded in NASA's human factors and behavioral research [9–11], and how simulating the complexities of team dynamics can have an operational impact on space missions.

## Materials and methods

### Complex systems and agent-based models

The Artemis program meets the definition of a complex system: an ensemble of many elements interacting in a decentralized order, forming hierarchical structures

and retaining memory [12]. More specifically, the Lunar Base can be considered a complex adaptive system – a subset of complex systems that involves adaptation to exogenous and endogenous factors, which may result in emergent properties. Although similar to chaotic systems, which are highly sensitive to initial conditions, the Lunar Base also includes effects of feedback loops and the potential emergence of self-organizing behaviors that are the result of multiple interactions between agents and their extreme environments [13].

The Lunar Base can be considered as an interesting case where top-down (highly and minutely designed systems) and bottom-up (agentized interactions and self-organized, spontaneous systems) phenomena intersect, and lead to unpredictable emergent consequences in space. The top-down layer includes highly structured mission planning, engineering design, and resource allocation that are fixed, formalized, and centrally imposed. In contrast, the bottom-up layer emerges from local agent interactions involving astronauts collaborating, coping with psychological pressure, and addressing unexpected tasks from exogenous shocks in a decentralized manner. It is the intersection between these two complex phenomena that can produce emergent dynamics, which can escalate into mission-level failures, including loss of life and resources. These unpredictable emergent consequences can potentially become the point of failure of a mission, with catastrophic results. It is critical to understand and model this dynamic for understanding resilience and contingency ahead of future space mission launches, after which the margin of error can only be minimal.

ABMs are insightful and effective tools for understanding the dynamics of complex systems that involve many interactions between heterogeneous actors and the flow of information, energy, and resources [8]. ABMs capture the effects of diverse agents, multiple complex interactions between agents and the environment, non-equilibrium states, and both positive and negative feedback loops. In our model, astronauts and autonomous rovers, each with individual preferences, strategies, and behaviors, represent agents. The ABM environment reflects the lunar geography, resources, and other factors (e.g., gravity, temperature, radiation, solar wind, regolith, moonquakes, and impact gardening) that can limit or promote agent interactions.

## Integrated decision-making framework inspired by Agent_Zero

In developing the design architecture of the Astronaut agents, Epstein's Agent_Zero framework [14] was used as a starting point, which models decision-making as the sum of three interacting components: cognitive (deliberative), affective (emotional), and social (peer influence). This new conceptual entity that possesses affective, cognitive, and social characteristics, interact with other similar entities and its environment to determine its observed behavior. As a result, Agent_Zero can be simulated and analyzed in an ABM. The total disposition of the Agent_Zero model for Agent $i$ can be summarized by the following skeletal equation:

$$D_i^{tot}(t) = V_i(t) + P_i(t) + \sum_{j \neq i} \omega_{ji} \left( V_j(t) + P_j(t) \right) .$$

(1)

The components of Eq 1 include the affective/emotional function, $V_i(t)$, the cognitive/deliberative function, $P_i(t)$, and the weighted solo dispositions of all other agents, $\omega_{ji}(V_j(t) + P_j(t))$. If the total disposition, $D_i^{tot}(t)$, for Agent $i$ exceeds a threshold, $\tau$, at any time, then a specified action is taken. Otherwise, there is no action. In his book, Epstein applies the Agent_Zero model to study the fight-or-flight dynamics of human behavior and other complex social phenomena, illustrating how emotional influence and homophily, or the tendency for individuals to align with others who share similar feelings, can shape collective patterns.

While the Lunar Base ABM retains the high-level tripartite structure of affective, cognitive, and social factors, the model diverges in three key ways from the Agent_Zero framework. First, Epstein's model was designed to capture fear-based behavior of agents, whereas, the Lunar Base ABM primary agents (i.e., astronauts) differ fundamentally since they are not emotionally blank slates – they are highly trained individuals with professional skills, distinct personalities, prior

socialization, and adaptive psychological traits who were hand-picked by space agencies to perform complex deep space missions. Second, the behavioral dispositions of the Astronaut agents are modeled as multiplicative instead of additive involving functions of emotional, professional, and learning rates that change over time. This nonlinear structure captures the compounding effects of psychological health degradation under stress and professional skill learning improvement over time taking into account personality traits and extreme environmental factors. Third, while Agent_Zero modeled emotion through the Rescorla-Wagner rule (i.e., the conditioning of an agent's emotional response based on the discrepancy between expected and actual outcomes over time), the Lunar Base ABM captures the affect via using coping capacity and tension interrelationship rules that can be validated using analogous space mission data. The integration of affective, cognitive, and social factors results in a new Agent_Astronaut framework tailored to the demands of space crewed missions and serves as the foundation for the Lunar Base ABM.

### Psychological adaptation to extreme environments

In preparation for future space missions, NASA has funded research related to human psychological adaptation to extreme environments. One joint study, funded by the NASA Office of Life and Microgravity Sciences and the Australian Antarctic Division, involved a team of three scientists and three engineers traversing the Lambert Glacier Basin from 1993 to 1994 and 1994–1995 [15]. The purpose of the study is to understand psychological responses in an extreme environment that is analogous to a NASA planetary space mission. The expedition lasted 110 days and involved 3 heavy bulldozers pulling sleds that carried habitation, generators, and laboratory vans with the relevant supplies for a total distance of 2,250 km. The participants in the study completed self-reported questionnaires regularly, which consisted of items covering emotional states, performance, and environmental factors. In the questionnaire, subjects were asked to rate their experiences related to the following measures: group tensions, individual morale, emotional state, cognitive readiness, and teamwork life. The Lunar Base ABM utilizes the study's results to compare and validate the coping capacity (i.e., individual psychological health, encompassing mood, depression, motivation, irritability, etc.) and tension level (i.e., interpersonal feelings among team members) as the affective function for Agent_Astronaut.

### DISC personality types

Tension between team members and the ability to work together to perform tasks are significantly influenced by their personality types. One way to characterize individual personality is the DISC method [16], which categorizes individuals into four personality styles: dominant, influencer, steady, and conscientious. The dominant personality types are those who make decisions quickly and take control of the situation. They are direct in their actions and unafraid of confrontation. The influencers are individuals who are people-oriented and excel in communication and motivating others. The steady or stable types are loyal, team-oriented, and want to maintain equilibrium. They are also the most common personality type. The final personality style is conscientious, who are task-oriented and analytical or precise with their actions. Although any personality type can work together effectively, certain combinations create more tension than others [17]. In general, dominant and conscientious types work well together because the direct and analytical approaches complement each other. Influencers and steady types are also effective in creating a collaborative and team-oriented environment. An example of a challenging combination of personality types is that of dominant and steady individuals, where the former are quick to establish cultural change to address a problem, whereas the latter prefer to maintain the status quo.

### NASA space mission skills

The necessary cognitive or deliberative skills for the Lunar Base ABM can be identified through NASA space mission experiences [18]. The average workday for an astronaut onboard the space shuttle (STS) or ISS has been approximately 16 hours plus overtime. Shuttle and station maintenance have been daily occurrences that consume a significant portion of the day and require strong engineering skills to ensure that life support and operational equipment are functioning

properly. In addition to routine maintenance and troubleshooting for equipment failure, crew activities include assembling activities (e.g., upgrading or adding new modules on the ISS) to increase capability for future space missions. Flight operations, planning, and daily conferences with Mission Control ensure situational awareness of the key events of the day and highlight the significant risks that the crew faces. Furthermore, the primary objective of the ISS is to conduct research that requires some Astronauts to possess a scientific background to conduct experiments and advance human knowledge. Lastly, each astronaut must be trained in extravehicular activity (EVA) operations to conduct spacewalks necessary for maintaining station infrastructure or conducting scientific exploration.

## Learning rate

The technology or experience learning curve is a common manufacturing concept that involves performing repetitive and labor-intensive tasks. According to the 2008 NASA Cost Estimating Handbook, "the learning curve effect states that the more times a task has been performed, the less time will be required on each subsequent iteration… The major premise of learning curves is that each time the product quantity doubles, resources (labor hours) required to produce the product will reduce by a determined percentage [slope] of the prior quantity resource requirements." [19] The unit value (i.e., labor time or cost) is calculated according to the following equation for a specific point on the curve:

$$Y = AX^b$$

(2)

$Y$ = Unit value of the $X$th unit
$A$ = Theoretical first unit value (T1)
$X$ = Unit number
$b = \dfrac{\log(\text{slope})}{\log(2)}$
slope = Percentage of reduction in unit resource cost each time production doubles

Table 1 presents the typical learning curve slopes by industry for benchmarking purposes.

## NASA TLX methodology

To assess the workload on its missions, NASA developed the Task Load Index (TLX), a standardized method for measuring task performance. According to NASA, "workload is defined as the cost incurred by a human operator to achieve a particular level of performance… workload is not an inherent property, but rather it emerges from the interaction between the requirements of a task, the circumstances under which it is performed, and the skill, behaviors, and perceptions of the

**Table 1. Slope by industry [19].**

| Industry | Slope (percentage) |
| --- | --- |
| Aerospace | 85% |
| Electronics manufacturing | 90–95% |
| Electrical operations | 75–85% |
| Raw materials | 93–96% |
| Complex machine tools | 75–85% |
| Machining or punch press | 90–95% |
| Welding operations | 90% |
| Purchased parts | 85–88% |

operator." [20] The conceptual framework illustrating the relationship between workload performance and operator behavior is presented in Fig 1.

To determine the TLX after task performance, participants rate six subscales, each ranging from 0 to 100, consisting of mental demand, physical demand, temporal demand, performance, effort, and frustration level. The total workload score is calculated by the weighted average of the six subscales. The NASA TLX methodology has been used to assess performance across various tasks, including cognitive, physical, supervisory, and operational tasks, in multiple industries such as space, military, and medical fields. Consequently, the Lunar Base ABM will use TLX data from analogous research projects to calibrate model output and validate task performance.

## Methodology and ABM overview, design concepts, and details

**Key assumptions.** The following assumptions were made for the initial case of the Lunar Base ABM:

1. Moon Base and Gateway are already constructed and in operations. Life support systems, power generators, local resource production (i.e., air, water, food), and other essential infrastructure are in place to support ongoing missions.

2. Nuclear generators (e.g., Perseverance power source – the first automated rover currently operating on Mars Jezero crater to use nuclear power instead of solar power, with an estimated energy life of 14 Earth years that is not impacted

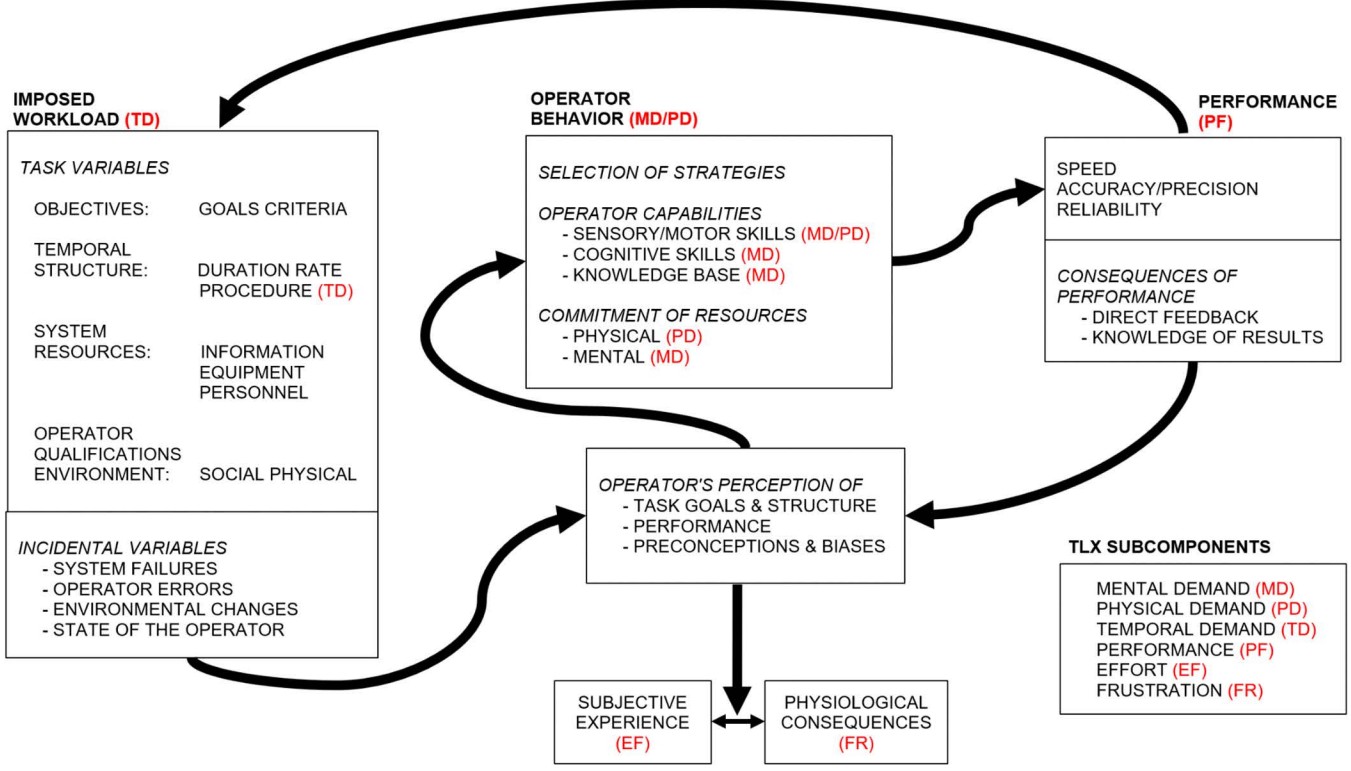

**Fig 1. Conceptual framework for relating human performance and workload.** From Hart & Staveland (1988) [20], this model illustrates the relationship between imposed workload, operator behavior, and resulting performance. It accounts for task demands, operator strategies, perception, and subjective/physiological consequences. The figure also shows how the TLX subcomponents (i.e., mental demand, physical demand, temporal demand, performance, effort, and frustration) are integrated in the conceptual framework.

by surface conditions such as dust storms) are installed in the Moon Base and have provided a steady source of electricity for several years [21].

3. Communication systems between Earth (e.g., Mission Control), the Moon Base, and the Gateway are operational, providing clear and consistent situational awareness among all three locations.

**Problem formulation.** The main objective of the model is to simulate a theoretical lunar mission environment, including both the surface habitat (Moon Base) and the orbiting Gateway station, for astronauts to perform relevant space mission tasks. The successful completion of the mission is measured by task performance, which is significantly influenced by cognitive skills, psychological state, and interpersonal relationships, in addition to the exogenous factors of the extreme environment.

**Model description.** The purpose of the Lunar Base ABM is to provide insight into the human and operational dynamics in extreme environments. The approach utilizes NASA's human factors and behavioral research, which emphasizes teamwork and psychological resilience, to simulate how astronauts can interact with one another to accomplish tasks and sustain operations on the Moon. Each Agent_Astronaut is designed with a set of attributes, including health, skills, and personality traits, that enable mission planners to simulate the complexities of team dynamics. The lunar system also increases in complexity as the number of agents, the variability of endogenous and exogenous parameters, and the mission duration increase. The goal of the Lunar Base ABM is to address complex systems dynamics in space by simulating human factors and interactions relevant to the Artemis missions in a timely and efficient manner.

The ABM was developed in Python and Mesa [22] and consists of one primary agent type (i.e., Agent_Astronaut) and six supporting agent types: Rover, Moon Base, Gateway, Shuttle, Moon Lander, and Task. Fig 2 shows the class diagram for the ABM of the seven classes of agents, with associated attributes. Each class consists of a unique identifier. Astronaut agents possess physical health, coping capacity, tension level, and professional skills (i.e., EVA, science, engineering, and flight operations) and exhibit one of the four DISC personality types (i.e., dominant, influential, steady, and conscientious). When traversing the lunar landscape, rovers have energy meters, cargo capacity, and increased mobility. The Moon Base and Gateway have an energy meter and keep track of the amount of air, water, food, and waste storage. Each Agent_Astronaut is assigned to only one Moon Base or Gateway agent. The task agents consist of a type (i.e., exploration, science, assembly, maintenance, troubleshooting, mission planning, and infrastructure) and skill requirements that are necessary for one astronaut (or a team of astronauts) to achieve in order to accomplish the activity.

Astronauts can pair up to perform tasks (resource production, exploration, science, assembly, maintenance, troubleshooting, and mission planning) inside and outside the Moon Base, consume resources (air, water, and food), and produce waste. Rovers can operate outside on the surface of the Moon to perform exploration and infrastructure activities while consuming energy. They recharge and undergo routine maintenance daily while docking at the Moon Base. In addition to transporting new astronauts to the Moon, shuttles perform resupply missions of air, water, and food every two months to ensure that crew members do not run out of resources and die.

Tables 2-8 list the attributes and starting values for the Lunar Base ABM agents or classes.

The temporal scale of the Lunar Base ABM is that one step is equivalent to one Earth hour. For example, a 3-month space mission will equate to 2,016 hours of duration since the model schedule operates in 24-hour cycles per day. Moreover, the spatial scale is one grid square in the multi-grid output, equivalent to approximately 1 square kilometer. The exception to this rule is that the Gateway and its associated astronauts are not assigned physical locations within the multi-grid framework, as they operate in orbit around the Moon.

**Process overview and scheduling.** The user input into the model is the number of astronauts, autonomous rovers, size of the lunar landscape (square km), total simulation duration (hours), resource shipment frequency (hours), astronaut replacement and transfer frequencies (hours), and exogenous factor probabilities (i.e., radiation, impact gardening,

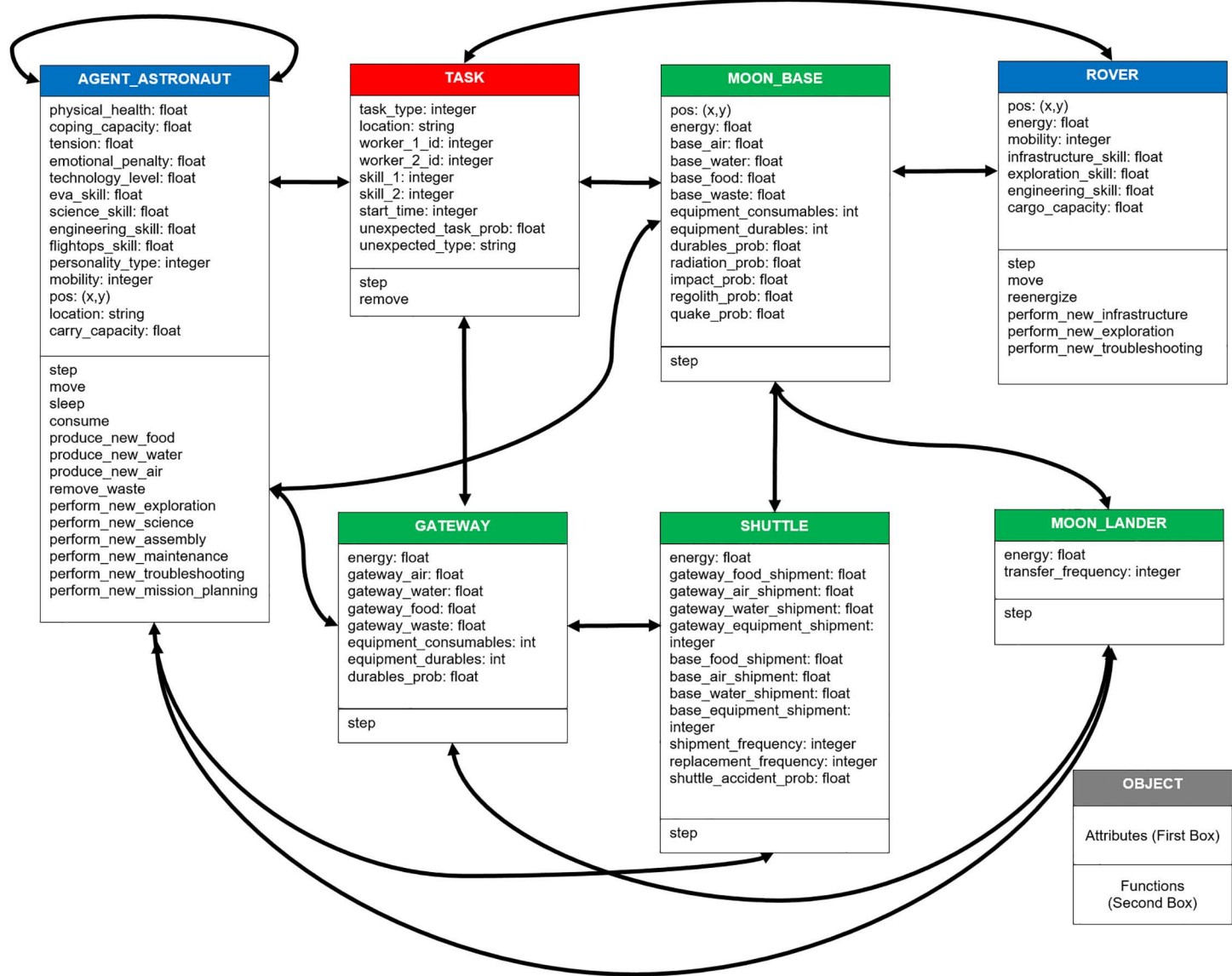

**Fig 2. Lunar Base ABM class diagram.** Class diagram showing the main agent and object classes used in the simulation: Agent_Astronaut, Rover, Task, Moon Base, Gateway, Shuttle, and Moon Lander. Arrows indicate interactions between agents, such as task assignment and resource flows. Object classes consist of main attributes (first box) and functions (second box). This diagram reflects the structural architecture of the agent-based model developed using Python and Mesa.

regolith, moonquake, durables, shuttle accident, and unexpected task). The steps of the Lunar Base ABM by the time of Earth Day (hour) are listed in Table 9 and follow the general schedule of past human space missions [23,24].

**Modeling Agent_Astronaut psychology and performance.** Astronaut performance in long-duration space missions is shaped not only by technical skill but also by psychological resilience, interpersonal relationships, and adaptability under stress. To represent these human dimensions in the Lunar Base ABM, four key components are integrated that are informed by NASA studies and analogous environments: cognitive professional skills, psychological health (i.e., combination of coping capacity and tension), DISC personality types, and learning or experience rates. Compared to the

**Table 2. Agent_Astronaut attributes.**

| Attribute Name | Description | Starting Value(s) |
|---|---|---|
| physical_health | physical health value | 1.0 |
| coping_capacity | personal psychological health factor | 1.0 |
| tension | interpersonal psychological factor | 0.0 |
| eva_skill | professional skill for EVA and exploration | 0.0–1.0 |
| science_skill | professional skill for science | 0.0–1.0 |
| engineering_skill | professional skill for engineering (maintenance, assembly, troubleshooting, resource production) | 0.0–1.0 |
| flightops_skill | professional skill for flight operations and mission planning | 0.0–1.0 |
| personality_type | DISC personality type | 1 = dominant, 2 = influential, 3 = steady, 4 = conscientious |
| mobility | number of grid squares (1 km) an astronaut can travel in 1 hour (step) | 1 |
| pos | location on the Moon's surface | (x,y) |
| carry_capacity | carry capacity while conducting EVA operations (kilograms (kg)) | 10.0 |
| technology_level | technology/experience efficiency | 0.5 |
| learning_rate | learning curve rate for technology_level | 0.85 |

**Table 3. Rover (autonomous) agent attributes.**

| Attribute Name | Description | Starting Value(s) |
|---|---|---|
| energy | energy value (kWh) | 1.0 |
| mobility | number of grid squares (1 km) rover can travel in 1 hour (step) | 10 |
| infrastructure_skill | professional skill for infrastructure | 0.0–1.0 |
| exploration_skill | professional skill for exploration | 0.0–1.0 |
| engineering_skill | professional skill for engineering | 0.0–1.0 |
| cargo_capacity | cargo capacity (kg) | 800.0 |
| pos | physical location of rover | (x,y) |

Agent_Zero model that was used as a starting point, the professional skills are analogous to the cognitive component, psychological health is similar to the affective component, and the combination of the personality types with teamwork dynamics capture the social component. The learning rate simulates how the skill rates change over time due to experience. Together, these attributes interact to determine the probability of task completion success, individual health, team cohesion, and mission sustainability.

Astronauts are randomly assigned DISC personality types (i.e., dominant, influential, steady, or conscientious) using a uniform distribution based on the lack of available data. Future research includes collecting historical personality type data on astronauts to develop a distribution that provides a more reasonable estimate of actual behavior. Moreover, the DISC personality type combinations influence the psychological cost or benefit when working in teams [17]. For instance, the dominant and steady tend to exhibit interpersonal friction, whereas, dominant-conscientious and steady-influential pairs are more compatible. These individual traits affect team interactions in addition to individual astronaut coping capacity and tension levels that result in emotional benefits or penalties that impact task outcomes.

**Table 4. Moon Base agent attributes.**

| Attribute Name | Description | Starting Value(s) |
|---|---|---|
| pos | physical location of Moon Base | (x,y) |
| energy | energy value (kWh) | 6000.0 |
| base_air | air supply (kg) | 180-day supply * num of Astro * 0.895 |
| base_water | water supply (kg) | 180-day supply * num of Astro * 5.03 |
| base_food | food supply (kg) | 180-day supply * num of Astro * 2.39 |
| base_waste | accumulated waste (kg) | 0.0 |
| radiation | daily probability for radiation event | 0.005–0.024 |
| impact_gardening | daily probability for impact gardening event | 0.80 |
| regolith | daily probability of regolith occurrence | 0.05–0.17 |
| moon_quake | daily probability of moonquake event | 0.00096 |
| equipment_consumables | equipment stockpile for maintenance and repair | 100 |
| equipment_durables | durable items for maintenance and repair | 20 |
| durables_prob | daily probability of using up durable | 0.0001–0.0008 |
| unexpected_task_prob | daily probability of unexpected task | 0.0 - 1.0 |

**Table 5. Gateway agent attributes.**

| Attribute Name | Description | Starting Value(s) |
|---|---|---|
| energy | energy value (kWh) | 4500.0 |
| gateway_air | air supply (kg) | 180-day supply * num of Astro * 0.895 |
| gateway_water | water supply (kg) | 180-day supply * num of Astro * 5.03 |
| gateway_food | food supply (kg) | 180-day supply * num of Astro * 2.39 |
| gateway_waste | accumulated waste (kg) | 0.0 |
| equipment_consumables | equipment stockpile for maintenance and repair | 100 |
| equipment_durables | durable items for maintenance and repair | 20 |
| durables_prob | daily probability of using up durable | 0.0001–0.0008 |
| unexpected_task_prob | daily probability of unexpected task | 0.0 - 1.0 |

Drawing from NASA operational experience [18], Astronaut agents are randomly initialized with professional skill ratings in EVA, science, engineering, and flight operations from a uniform distribution. These values represent the functional capacity to complete specific tasks according to the mission schedule. For example, EVA skills are used for exploration tasks and engineering skills are utilized for maintenance and assembly activities. Higher skill ratings increases the probability of successful task completion, either individually or in teamwork with others to achieve cumulative skill requirements.

Psychological health is modeled using two variables: coping capacity (i.e., internal emotional state) and tension (interpersonal strain). These factors change over time based on personality interactions, environmental stressors, and unexpected activities. Psychological health can further be tested and validated with psychological research from the Antarctic missions [15], where participants reported morale, emotional states, and team dynamics in a rigorous manner. The overall

**Table 6. Shuttle agent attributes.**

| Attribute Name | Description | Starting Value(s) |
|---|---|---|
| energy | energy value (kWh) | 4500.0 |
| gateway_air_shipment | 30-day gateway air resupply per astronaut (kg) | 30 * num of Astro * 0.895 |
| gateway_water_shipment | 30-day gateway water resupply per astronaut (kg) | 30 * num of Astro * 5.03 |
| gateway_food_shipment | 30-day gateway food resupply per astronaut (kg) | 30 * num of Astro * 2.39 |
| base_air_shipment | 30-day base air resupply per astronaut (kg) | 30 * num of Astro * 0.895 |
| base_water_shipment | 30-day base water resupply per astronaut (kg) | 30 * num of Astro * 5.03 |
| base_food_shipment | 30-day base food resupply per astronaut (kg) | 30 * num of Astro * 2.39 |
| shipment_frequency | frequency of supply shipments (hrs) | 1440 |
| base_replacement_frequency | frequency of base astronaut replacement (hrs) | 1440 |
| gateway_replacement_frequency | frequency of gateway astronaut replacement (hrs) | 1440 |
| shuttle_accident | probability of shuttle accident event per flight | 0.0645616 |

**Table 7. Moon lander agent attributes.**

| Attribute Name | Description | Starting Value(s) |
|---|---|---|
| energy | energy value (kWh) | 100.0 |
| transfer_frequency | frequency of transferring astronauts between base and gateway (hrs) | 0 |

**Table 8. Task agent attributes.**

| Attribute Name | Description | Starting Value(s) |
|---|---|---|
| task_type | task type | 1 = exploration, 2 = assembly, 3 = science, 4 = maintenance, 5 = troubleshooting, 6 = mission planning, 7 = infrastructure |
| skill_1 | professional skill 1 requirement | 0.0–1.0 |
| skill_2 | professional skill 2 requirement | 0.0–1.0 |
| unexpected_type | unexpected task description | ('normal', 'unexpected', 'radiation', 'impact', 'quake') |

stress from the combined coping capacity and tension levels can result in an emotional penalty that is applied multiplicatively to the task execution likelihood.

Skill effectiveness improves over time through repetitive experience using the inverse of the learning curve function [19]. Since the Lunar Base ABM tracks increase of capability rather than reduction of labor effort, the inverse curve represents the labor efficiency gained as astronauts repeat recurring tasks such as assembly or maintenance. The learning rate is initialized based on NASA benchmarks with a typical slope of 85%. This experience factor increases the effective disposition of tasks, compounding the cognitive and emotional variables over time, resulting in an overall probability increase of task success at the team level.

Together, these four key components (i.e., professional skills, coping capacity, tension level, and efficiency rate) offer a basis for simulating astronaut behavior under extreme environmental conditions. When integrated into the task disposition framework, they allow for realistic exploration of how personality types, professional skills, interpersonal dynamics, emotional resilience, and learning interact to shape mission success. The total disposition of performing an assigned activity

**Table 9. Daily astronaut schedule in the Lunar Base ABM.**

| Hour(s) | Activity Description |
|---|---|
| 1 | Astronauts wake up, exercise, and conduct personal hygiene. |
| 2 | Astronauts eat breakfast and conduct housecleaning. |
| 3 | Astronauts perform mission planning for the morning activities. If they meet the flight operation skill requirement, they are successful in performing the activity. They can also work together in pairs to achieve the necessary skill requirement if they are unable to do so individually. |
| 4 | Astronauts perform a series of base operations (assembly, science, maintenance, and troubleshooting) by meeting the relevant skill requirements of the tasks. They can also work together in pairs to achieve the necessary skill requirements. |
| 5 | Astronauts continue base operations from Hour 4. |
| 6 | Astronauts eat lunch and perform mission planning for the afternoon activities. They must meet the necessary flight operation skill requirement (solo or in pairs) to succeed in mission planning. |
| 7–12 | Astronauts perform EVA operations on the lunar surface, which include preparation time and habitat re-entry. Rovers also perform exploration and infrastructure activities outside the Moon Base at this time. Both types of agents must meet the skill requirements to be successful in their assigned tasks. |
| 13 | Astronauts eat dinner and perform mission debriefing for all of the day's activities. They must meet the necessary flight operation skill requirement (individually or in teams) to succeed. |
| 14 | Astronauts use their engineering skills to participate in resource production (air, water, and food) and waste management. |
| 15 | Astronauts experience personal time. |
| 16 | Astronauts exercise and conduct personal hygiene. |
| 17–24 | Astronauts sleep and rovers reenergize at the end of the workday. If the astronaut's physical health or the rover runs out of energy, they are removed from the simulation. |

for the primary worker, Agent_Astronaut $i$, with assistance from Agent_Astronaut $j$, can be summarized by the following equation:

$$D_i(t) = \left[ S_i \left( 1 - E_i(t) \right) + S_j \left( 1 - E_j(t) \right) \right] T(t)$$

(3)

The components of Eq 3 include the skill parameter (i.e., cognitive factor), $S_i$, emotional penalty (i.e., affect factor), $E_i(t)$, and the technology or experience factor $T(t)$. The skill parameter is a random number between 0.0 and 1.0, uniquely assigned to each astronaut. Emotional penalty is based on the combined effects of coping capacity and tension values that are equally weighted to produce the associated affect score. The technology factor is based on the inverse of the learning curve at time t, as specified in Eq 2, and a slope that is user-selected from the values in Table 1. The benefits of the technology or experience learning curve are only applicable to repetitive and labor-intensive activities, such as assembly, maintenance, and mission planning.

If the total disposition, $D_i(t)$, for Agent_Astronaut $i$ exceeds the activity $k$ threshold, $\tau_k$, then activity $k$ was successfully completed. Otherwise, the activity $k$ was not completed.

Specific professional skills and agent type are required to perform a particular activity. Table 10 defines the mapping between tasks, skills, and agent type in the Lunar Base ABM.

The Moon Base and Gateway receive resource resupply from the shuttle based on the resource shipment frequency. Astronauts are also transferred from Earth to the Moon and back again through the shuttle, depending on the frequency of astronaut replacement. The Lunar Base ABM has the option to transfer crew members between the Moon Base and the

**Table 10. Task skills mapping.**

| Task (t) | Professional Skill | Agent Type |
|---|---|---|
| Exploration (1) | EVA, Exploration | Astronaut, Rover |
| Assembly (2) | Engineering | Astronaut |
| Science (3) | Science | Astronaut |
| Maintenance (4) | Engineering | Astronaut |
| Troubleshooting (5) | Engineering | Astronaut |
| Mission Planning (6) | Flight Operations | Astronaut |
| Infrastructure (7) | Infrastructure | Rover |

Gateway via the Moon lander, utilizing the astronaut transfer frequency parameter. There is also a small probability that the shuttle will experience an accident that will result in the loss of astronauts and resources.

Astronauts encounter unexpected tasks throughout the day that they must address based on user-defined parameters. They can also encounter environmental events that include radiation, impact gardening, and moonquakes. For unexpected activities, the astronauts must deviate from the daily schedule and perform troubleshooting to fix any equipment malfunctions that have occurred due to the exogenous event.

Fig 3 is the Lunar Base ABM input-output flow diagram that summarizes the exogenous and endogenous inputs from the left and the top/bottom of the box, respectively, as well as the corresponding model output on the right side of the box. The exogenous input variables represent environmental factors (e.g., shuttle accident probability, radiation probability), while the endogenous input variables account for internal factors inherent to the primary Agent_Astronaut (e.g., personality type, EVA skill). The main output parameters of the Lunar Base ABM include the synthetic TLX score, astronaut coping capacity/tension, total task completion, and resource production values.

### Design concept.

- *Basic principles.* The Lunar Base ABM integrates concepts from complexity science, human psychological factors, and agent-based modeling to understand the social and operational dynamics of a human base in an extreme environment, in this case the Moon. Each astronaut and rover operates autonomously but also interacts with each other and contributes to the overall space mission by accomplishing necessary tasks and producing or consuming resources. The Lunar Base ABM utilizes the Agent_Zero model for decision-making and task accomplishment, incorporating professional skills, personal and interpersonal emotional factors, and learning based on the performance of repetitive tasks. Through the interactions between agents and the environment, the model can be used to benchmark performance aligned with NASA's TLX methodology while accounting for the psychological stress that accompanies working in extreme environments.

- *Emergence.* Emergent phenomena in the Lunar Base ABM include fluctuations and distributions of the synthetic TLX score, along with corresponding cumulative mission success rates, coping capacity values, and tension levels. These output parameters are influenced by the heterogeneity of the astronauts' professional skill levels, personality types, and their interactions with one another. Space environmental factors, equipment failure, and accidents also play a significant role in determining the outcome of the synthetic TLX scores and the overall mission success.

- *Adaptation.* Agents adapt dynamically to environmental stressors and interpersonal team conditions. Depending on their professional skill level, astronauts are optimally chosen to perform a specific task and will seek partnership if they cannot complete the activity alone. Astronauts also consume resources, exercise, engage in personal time, and sleep to maintain and improve their physical and psychological health. Similarly, autonomous rovers will recharge overnight and undergo daily maintenance to be ready for the next day's activities. Extreme environmental

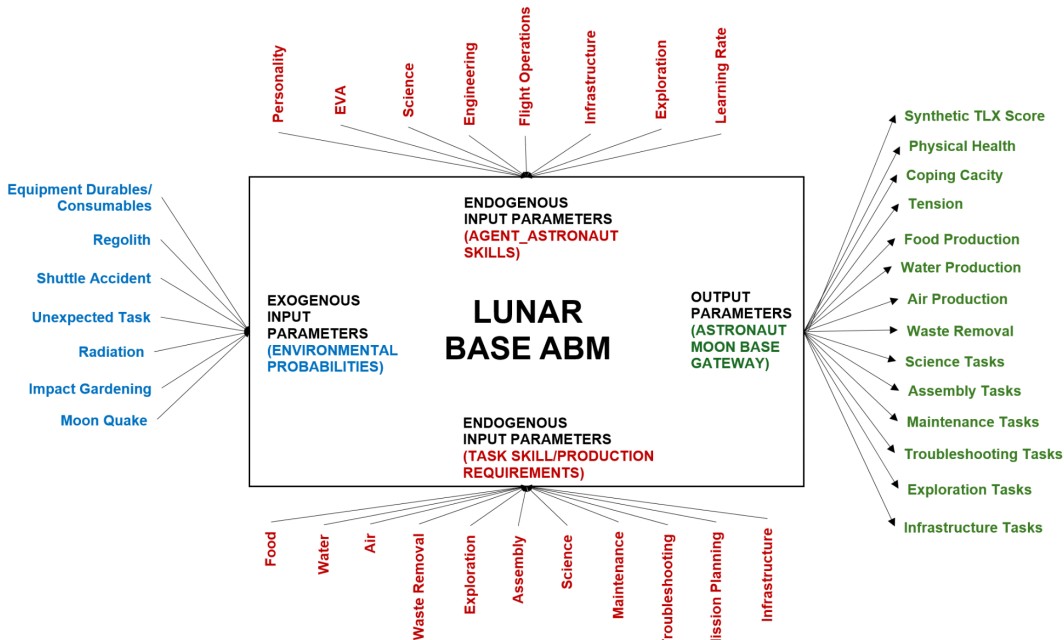

**Fig 3. Lunar Base ABM input-output flow diagram.** This diagram illustrates the mapping between exogenous parameters (left, in blue), endogenous astronaut and task-related parameters (top and bottom, in red), and the model output indicators (right, in green). The flow of information represents how simulation inputs are processed to generate key performance metrics such as TLX score, coping capacity, tension, and task completion.

events, such as radiation, solar wind, or moonquakes, will trigger adaptive behavior in the crew, who will pair with the best engineering skills and initiate emergency protocols, including returning to base. Moreover, the model incorporates learning by incrementally updating the overall experience or technology level based on the number of successful task completions.

- *Objectives.* The goal of the Lunar Base ABM is to simulate a typical lunar space mission involving a team of astronauts working together to complete daily activities while overcoming personality differences, managing resources, and dealing with extreme environmental events. Astronauts aim to complete assigned tasks, maintain physical and emotional well-being, and sustain the lunar habitat through daily maintenance and resource production. Rovers' primary objective is to conduct exploration and infrastructure-related missions. Shuttles and Moon landers serve as support systems, ensuring that astronauts are transferred to the appropriate location and have the necessary supplies to survive and complete their mission.

- *Sensing.* While conducting EVA operations on the lunar surface, astronauts can move up to one grid square (or one km) in one hour to one of its von Neumann neighbor grid squares (i.e., north, south, east, and west). During this time, they conduct exploration or science tasks alone or in pairs based on their skill rating. Autonomous rovers can move up to 10 grid squares per hour to perform exploration and infrastructure-building activities. All agents depart the habitat and return based on the daily schedule. In unexpected events involving environmental stressors (e.g., moonquakes, impact gardening, etc.), agents will stop what they are doing, troubleshoot, and return to base to address any equipment failures.

- *Interaction.* Astronaut agents interact with each other and the environment to complete tasks and sustain the Moon Base or Gateway. In general, when different personality types work together to perform activities, the agent incurs a

penalty for conflicting styles and a benefit for complementary styles in accordance with previous literature [17]. Specifically in the model, dominant-steady and influential-conscientious combinations will experience a 0.05 reduction in coping capacity and a 0.05 increase in tension level. In contrast, dominant-conscientious and influential-steady combinations will take a 0.05 increase in coping capacity and a 0.05 decrease in tension level. When astronauts interact with the environment, they will experience a 0.05 reduction in their coping capacity for unexpected events and environmental stressors. These coping capacity and tension level adjustments are based on the underlining mechanics of the NASA TLX test [20] and can be refined with new data from future research.

- *Stochasticity.* There are endogenous and exogenous sources for stochasticity in the Lunar Base ABM. Internal stochastic parameters include the astronauts' professional skill ratings (e.g., engineering, flight operations) and task skill requirements (e.g., resource production, assembly, maintenance). External stochastic parameters include environmental stressor probabilities, shuttle accident probability, and the frequency of durable tool usage. ABM's random variable distributions are either uniform between the minimum and maximum values or triangular, with minimum, maximum, and most likely values.

- *Collectives.* Astronaut agents form a collective to perform mission tasks, produce resources, deal with unexpected events, and sustain the Moon Base or Gateway habitat. Resources are pooled together in the Moon Base or Gateway for astronauts to consume and are replenished with frequent resupply missions from the shuttle. Waste management is also an essential habitat operation that the astronauts perform daily to ensure sustainability. If the agents produce excess resources (i.e., air, water, food) or waste, they are added to the common reserves or storage.

- *Observation.* The model outputs include synthetic TLX scores, astronaut physical health, coping capacity, tension values, habitat resource production, waste removal, equipment consumption, and durables usage, as well as mission task completion. The default simulation time is three Earth months, which can be adjusted as necessary.

   **Initialization.**  Two Python files create all the classes for the Mesa ABM architecture: Agents.py and Lunar_Base_ABM.py. Agents.py generates all the agent objects and associated attributes, including the Agent_Astronaut, Rover, Moon_Base, Gateway, Shuttle, and Moon_Lander agents. Lunar_Base_ABM.py initializes the overall Moon model object class, global input variables, model scheduler, and reporter variables that are included in the output files. At the start of each simulation run, astronauts are randomly assigned to the Moon Base or Gateway with randomly generated skill attributes and randomly assigned personality types. Environmental parameters such as radiation, moonquakes, and regolith obstruction are established with user-defined probabilities. No input data is used in this model.

   **Setting parameters.**  ABMs act as virtual laboratories for physical and social scenarios that cannot be replicated in the current environment and technology. As a result, one of the significant challenges with ABMs is setting the initial parameters to reasonable values while accounting for uncertainty. For the Lunar Base ABM, potential data sources for determining initial values include data from historic space missions, analogies involving extreme environments (e.g., Antarctica habitats), and observational data collected from scientific instruments (e.g., Lunar Reconnaissance Orbiter Camera (LROC)).

   The Lunar Base ABM global variables include the total simulation duration, the number of astronauts, the number of rovers, and the total grid area that represents the lunar surface. One NASA Artemis planning article accounts for a 30- to 60-day Artemis habitation period for a crew of up to four astronauts, which will eventually expand for further exploration on the lunar surface [25]. Given the fact that the typical ISS mission lasts for 6 months, the range of the lunar space rotation should range between 2 and 6 months. As a result, the Lunar Base ABM will initialize with a 3-month simulation duration to allow for the development of interpersonal relationships among the two astronauts located in the Moon Base, the two astronauts operating the Gateway, and the two rovers that can be regularly maintained by the available astronauts and provide backup when one is inoperable. The article also states that the Artemis Base Camp will most likely be located

in the South Polar region of the Moon. This area features permanently shadowed craters that may contain ice water, which can be utilized for life support and fuel production. LROC topographic studies confirm that sunlit ridges around the Shackelton Crater, located in the South Pole, can provide enough light for solar power generation. Since the Shackleton Crater is 21 km in diameter, the Lunar Base ABM will be initialized with approximately double the size, or 50 square km, to account for additional infrastructure locations and exploration areas around the lunar landmark.

Agent-specific variables for the Agent_Astronaut, Rover, Moon Base, Gateway, Shuttle, and Moon Lander include mobility, resource production rates, resource initial supply values, resource resupply frequency, astronaut replacement frequency, and astronaut transfer frequency between the Moon Base and Gateway. Historical Apollo missions found that the EVA speed for astronauts ranged between 1.1 kilometers per hour (km/h) and 2.2 km/h [26]. Additionally, the lunar rover vehicle (LRV) in the Apollo missions had a maximum speed of 18.0 km/h. Consequently, the Lunar Base ABM will use conservative values for EVA and rover speed of 1 km/h and 10 km/h, respectively. For the daily resource consumption rates, the ISS had 2.39 kg of packaged food per person per day. NASA's Human Integration Design Handbook (HIDH) states an oxygen daily rate of 0.895 kg/day, with a contingency margin, and a daily water usage of 5.03 kg/day [27]. For resupply missions, the ISS received an average of 1 cargo mission every 60–90 days, including food, water, oxygen, and nitrogen. Suppose the max capacity of the historical ISS resupply missions varied from 2,500–6,000 kg (including scientific equipment, replacement hardware, and miscellaneous items). In that case, it is reasonable to initialize the Lunar Base ABM with 30 days of food, water, and air for resupply every 60 days or 1440 hours. Lastly, the model assumes there is no astronaut transfer between the Moon Base and Gateway throughout the 3-month mission for simplicity. However, this assumption can be relaxed in order to analyze different mission and team composition scenarios.

The exogenous factors that impact the model include probabilities for radiation events, impact gardening, moonquakes, shuttle accidents, regolith disruption, and durable equipment failure. Several studies have investigated solar particle events (SPEs), which involve the frequency of discharging large amounts of energetic particles that can cause significant harm to human health and damage to electronic equipment and scientific instruments [28]. The Lunar Base ABM will assume a low daily probability of 0.005 (1 SPE every 208 days), a high daily probability of 0.024 (1 every 42 days), and most likely daily probability of 0.012 (1 every 90 days). For an impact gardening event, the probability of puncture from a micrometeoroid can be derived from the data collected during the Lunar Orbiter program in the 1960s. The average rate of five Orbiters was 0.16 punctures per square meter per day [29]. Assuming an exposed surface area of 10 square meters and using a Poisson process distribution, the probability of at least one puncture in a day on the lunar infrastructure is approximately 0.80. Moreover, lunar seismic data were also collected during the Apollo missions from 1969 to 1977. During these 8 years, 28 moonquake events were identified, with body wave magnitudes ranging from 3.5 to 6.0 [30]. This equates to a daily moonquake probability of 0.00096 per day. To determine a shuttle accident probability, NASA's Commercial Resupply Services (CRS) provides data on the delivery of supplies to the ISS from commercial spacecraft (i.e., SpaceX Dragon and Orbital ATK Cygnus). During CRS Phase 1, there were 31 transport flights with two failures, resulting in a shuttle accident rate of 0.064516. Additionally, the regolith probability values can be calculated using experience with the EVA systems on the lunar surface during the Apollo missions [31]. From six Apollo missions, it was found that EVA suits were close to approaching failure after 21–75 hours of operating on the lunar surface due to scratched instruments, degraded radiator performance, and compromised seals. With an expected life of 336 hours for the EVA systems, 4 hours of daily exposure, and assuming a Poisson process distribution, the daily probability of failure due to regolith ranges from 0.05 to 0.17. Lastly, the probability of durable equipment failure is based on NASA ISS maintenance data for life support subsystems. For oxygen generation, carbon dioxide removal, urine processor assembly, and water processor assembly systems, the daily failure probabilities range from 0.0001 to 0.0008.

**Implementation verification.** For output verification and analysis, the Lunar Base ABM calculates a synthetic TLX score, which consists of a weighted average of the tension level mean, coping capacity mean, and total tasks completed, normalized from 0 to 1 with an additional scaling factor to adjust for the difficulty of space mission tasks. Agent_Astronaut

tension (i.e., interpersonal) and coping capacity (i.e., personal) output can be compared to the psychological data collected from the analogous Antarctica expedition across the Lambert Glacier Basin for validation and further calibration of the ABM. The total number of assigned tasks that were successfully completed serves as a measure of performance efficiency for benchmarking different model assumptions, scenarios, and team compositions. The overall synthetic TLX score can be validated and calibrated with TLX data collected from previous studies, including NASA simulated spaceflight [32], HERA project [33], EDEN ISS Greenhouse study [34,35], Indonesian Navy TLX study [36], and DIANA study [37]. The calibration of the synthetic TLX score to fit historical data can be calculated by the following equation:

$$x_i^{\text{adj}}(t) = \mu_2 + \left(\frac{\delta_2}{\delta_1}\right)(x_i(t) - \mu_1)$$

(4)

In Eq 4, the adjusted synthetic TLX score at time $t$ ($x_i^{\text{adj}}(t)$) is equal to the unadjusted TLX value at time $t$ ($x_i(t)$) minus the unadjusted mean ($\mu_1$) that is multiplied by the ratio of the new standard deviation over the unadjusted standard deviation $\left(\frac{\delta_2}{\delta_1}\right)$, and added to the new mean ($\mu_2$). Note that this calibration only changes the interpretation of the TLX values and does not change the fundamental simulation dynamics.

In addition to the temporal output of tension, coping capacity, task completion, and synthetic TLX score parameters, running a Monte Carlo simulation can help account for model uncertainty and develop probability distributions for the output variables. NASA employs the Monte Carlo simulation approach to calculate numerous scenarios by randomly selecting values for input variables, iterating approximately 10,000 times to develop probability distributions of cost and schedule outcomes for major acquisition projects [19]. The Lunar Base ABM uses a similar approach by selecting random values for personality type, professional skill level, task skill requirements, and environmental or exogenous probabilities. Instead of focusing on cost and schedule S-curves, the model will calculate cumulative probability distributions for task completion, tension level, coping capacity, and synthetic TLX score. Using this methodology, one can calculate confidence levels for different scenarios and quantify the impact of changes in input parameters.

## Results and discussion

### Initial case

The starting scenario for the Lunar Base ABM involves four astronauts and two rovers operating on the Moon Base and Gateway for a mission duration of 3 months, totaling 2,016 simulated hours. During this time, a resupply mission will occur at month 2 (i.e., 1,440 hours), involving a shipment of food, water, air, and new personnel. There will be no astronaut transfers between the Moon Base and Gateway. The initial exogenous parameters are set to the values discussed in the Setting parameters section, and the synthetic TLX score is calibrated to the EDEN ISS data using the methodology outlined in the Implementation verification section. A Monte Carlo simulation of 10,000 iterations was conducted to compare the ABM results to the benchmarks displayed in analogous historical studies. For comparison purposes, the benchmarks are presented as ranges to visualize the mean and standard deviation of the analogous data. The actual data distributions of the aforementioned data were not available at the time of this study, and an accurate comparison can be conducted in future research. Figs 4-6 show the histograms with fitted normal distributions for the output parameters of the Lunar Base ABM (i.e., total task completed, average coping capacity, and average tension).

For the Monte Carlo simulation of the initial case, the average total tasks completed, as shown in Fig 4, is 748.2 with a standard deviation of 72.8. There are approximately 3,780 scheduled and unexpected tasks during a 3-month lunar mission. The simulated range indicates a consistent team productivity rate of approximately 20% under baseline conditions with low variability and a coefficient of variation (CV) value of 0.10, which is acceptable for a typical manufacturing process. This productivity rate does not include the completion of unexpected tasks because they are outside the baseline of daily scheduled mission activities. As a result, mission planners can implement statistical process controls to monitor task

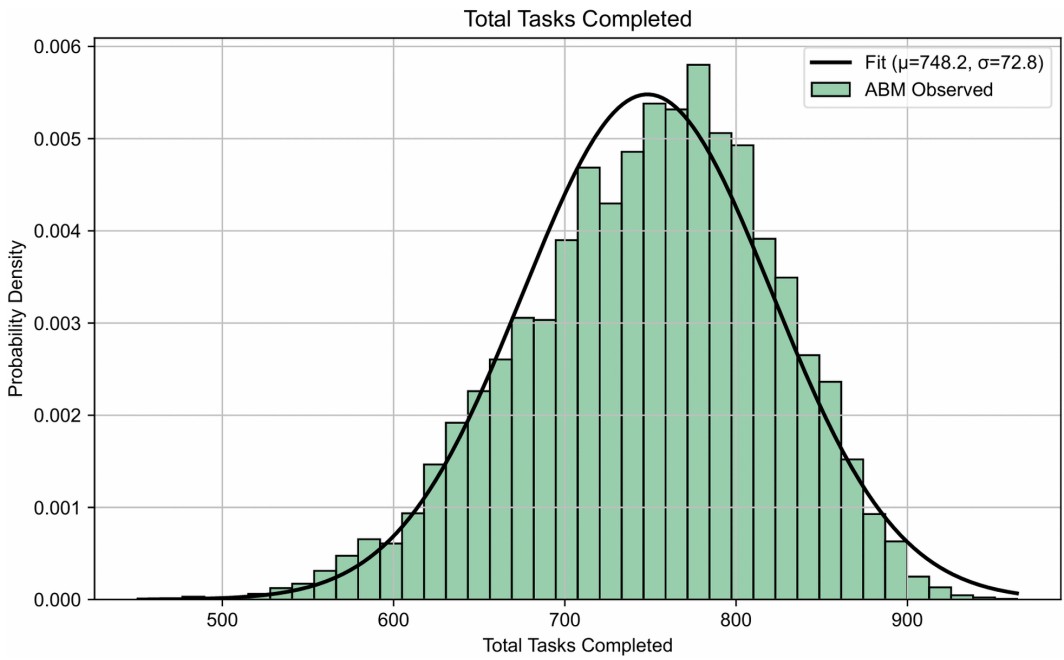

**Fig 4. Lunar Base ABM total tasks completed distribution (initial case).** Histogram and fitted distribution of total tasks completed during the simulation. The ABM results are compared to a normal fit with mean $\mu$ = 748.2 and standard deviation $\sigma$ = 72.8.

efficiency that may add value to future space missions. The low task completion rate suggests that, on average, teams are having challenges to overcoming psychological stressors and environmental disruptions in the initial configuration.

Fig 5 compares the average coping capacity output with the Antarctic Study 1993 and 1994 benchmarks [15] of individual moral and emotional states. The initial state output for coping capacity in the ABM aligns closely with the 1993 Antarctic data, with a mean of 0.67 and a standard deviation of 0.11, indicating that astronaut psychological health is well maintained during the simulated mission. This suggests that the initial team structure and stress load are within acceptable thresholds for psychological resilience. Futhermore, a kernel density estimation analysis reveals a single stable peak, confirming that the coping capacity output distribution is unimodal rather than multimodal.

Table 11 displays the mean and standard deviation values of the average coping capacity from the Lunar Base ABM and compares with the morale and emotional data values from the 1993 and 1994 Antarctic expeditions as shown in Fig 5.

In contrast, Fig 6 presents a comparison of the average tension level with the tension data collected during the 1993 and 1994 Antarctic expeditions. The interpersonal tension values from the model have a mean of 0.37 and a standard deviation of 0.20. This aligns with the 1993 and 1994 Antarctic Study benchmarks for tension indicating that astronaut interpersonal relationships are fairly represented in the Lunar Base ABM from the available historical data. Additional stressors involving personality traits or teamwork dynamics that are not being captured in the current ABM can be explored in future research. Although the histogram in Fig 6 shows a lower bound peak around 0.0, this is caused by the truncation of the ABM tension values to prevent negative values. Similar to the coping capacity output distribution, a smoothed kernel density estimation analysis shows only one stable peak indicating a unimodal distribution.

Table 12 displays the mean and standard deviation of the average tension level from the Lunar Base ABM and compares with the tension data from the 1993 and 1994 Antarctic expeditions as shown in Fig 6.

The combination of task completion, coping capacity, and tension, as described in the Implementation verification section, yields the synthetic TLX score distribution shown in Fig 7. The chart also shows a comparison of the TLX

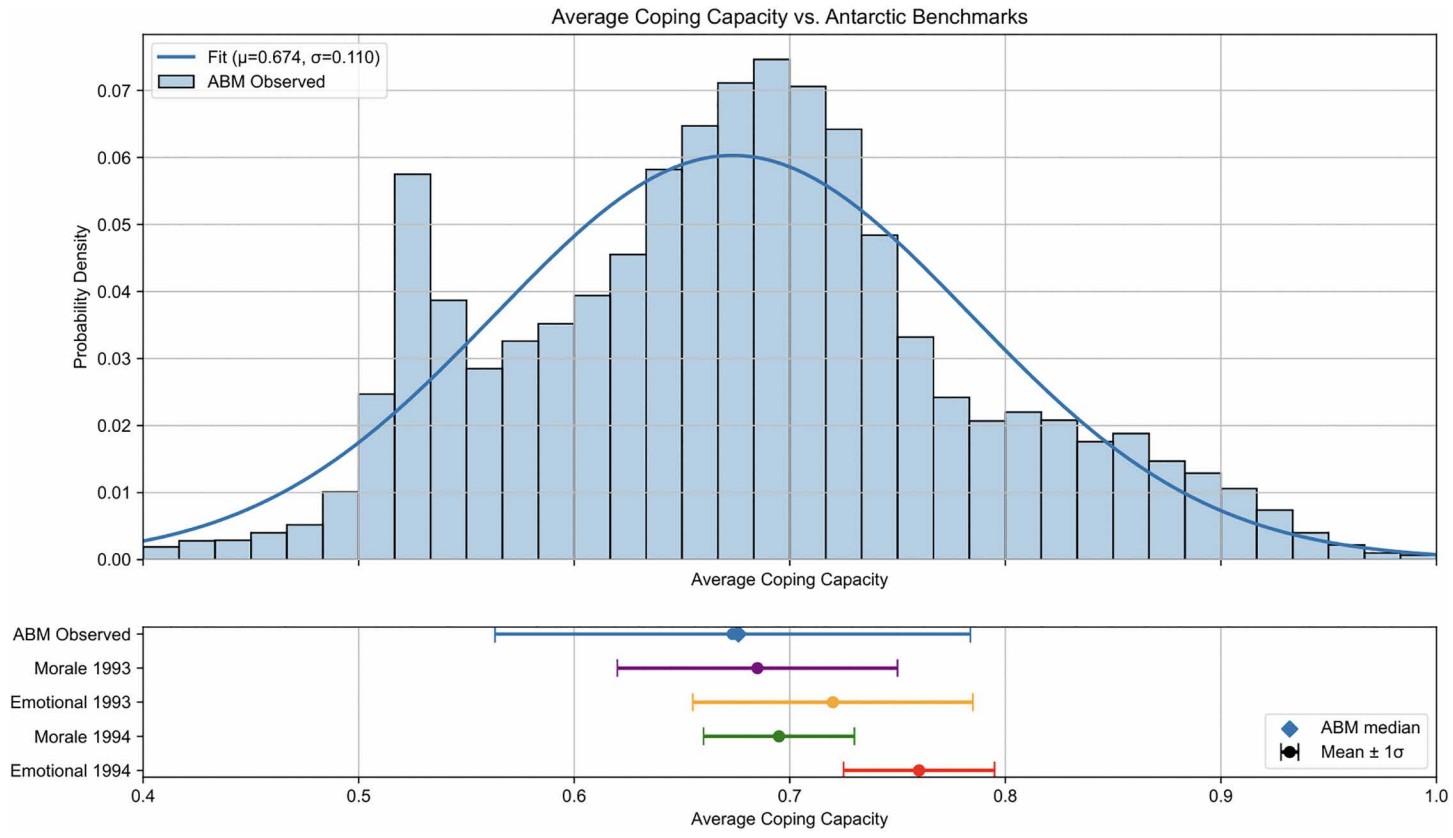

**Fig 5. Lunar Base ABM average coping capacity distribution (initial case).** Distribution of average coping capacity among astronauts compared against Antarctic expedition benchmarks for morale and emotional state from 1993 and 1994.

scores with historical analogous data [32–37]. As previously stated, the benchmarks are presented as ranges to visualize the mean and standard deviation of the analogous data because the complete datasets were not available at the time of this study. The calibration methodology in Eq 4 was used to adjust the synthetic TLX score to align it with the EDEN ISS study [34,35], which is the most analogous data to use for further analysis. The adjustment only alters the interpretation of the TLX values to examine the impact of input parameter changes without affecting the fundamental simulation dynamics. The simulated distribution demonstrates that most synthetic TLX scores fall within the middle to upper range of empirical benchmarks, suggesting moderate to high workload and mission effort. The model's calibration to EDEN ISS data reinforces validity in capturing the integrated cognitive, emotional, and environmental demands of lunar operations.

Table 13 displays the mean, standard deviation, minimum, maximum, and median of the synthetic TLX score from the Lunar Base ABM and compares with the TLX data from analogous studies as shown in Fig 7.

Fig 8 shows the cumulative distribution of the synthetic TLX score for the initial case after the Monte Carlo simulation. This cumulative view allows mission planners to estimate the likelihood of various workload levels occurring under baseline assumptions. For example, there is an 85% chance that TLX scores will remain below 72.55, a threshold consistent with the acceptable workload limit during analog missions. This metric serves as an indicator of the expected professional task effort and psychological impact of a lunar space mission based on initial conditions.

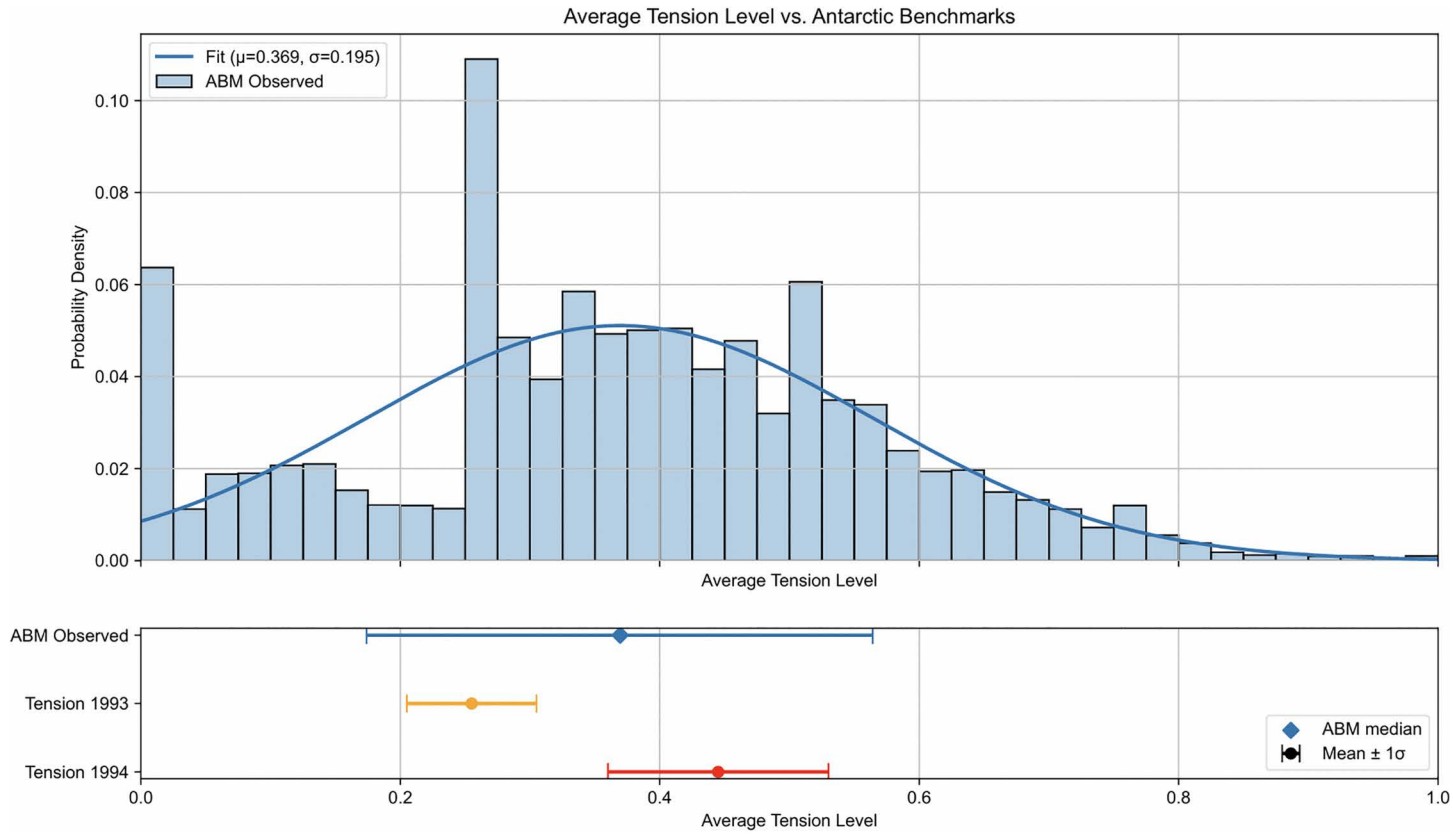

**Fig 6. Lunar Base ABM average tension level distribution (initial case).** Tension levels in the ABM are compared to Antarctic benchmarks from 1993 and 1994.

**Table 11. Lunar Base ABM average coping capacity comparison data.**

| Study Benchmark | Mean | Standard Deviation |
|---|---|---|
| Lunar Base ABM | 0.674 | 0.110 |
| Antarctic Morale 1993 | 0.685 | 0.065 |
| Antarctic Emotional 1993 | 0.720 | 0.065 |
| Antarctic Morale 1994 | 0.695 | 0.035 |
| Antarctic Emotional 1993 | 0.760 | 0.035 |

**Table 12. Lunar Base ABM average tension comparison data.**

| Study Benchmark | Mean | Standard Deviation |
|---|---|---|
| Lunar Base ABM | 0.369 | 0.195 |
| Antarctic Tension 1993 | 0.255 | 0.050 |
| Antarctic Tension 1994 | 0.445 | 0.085 |

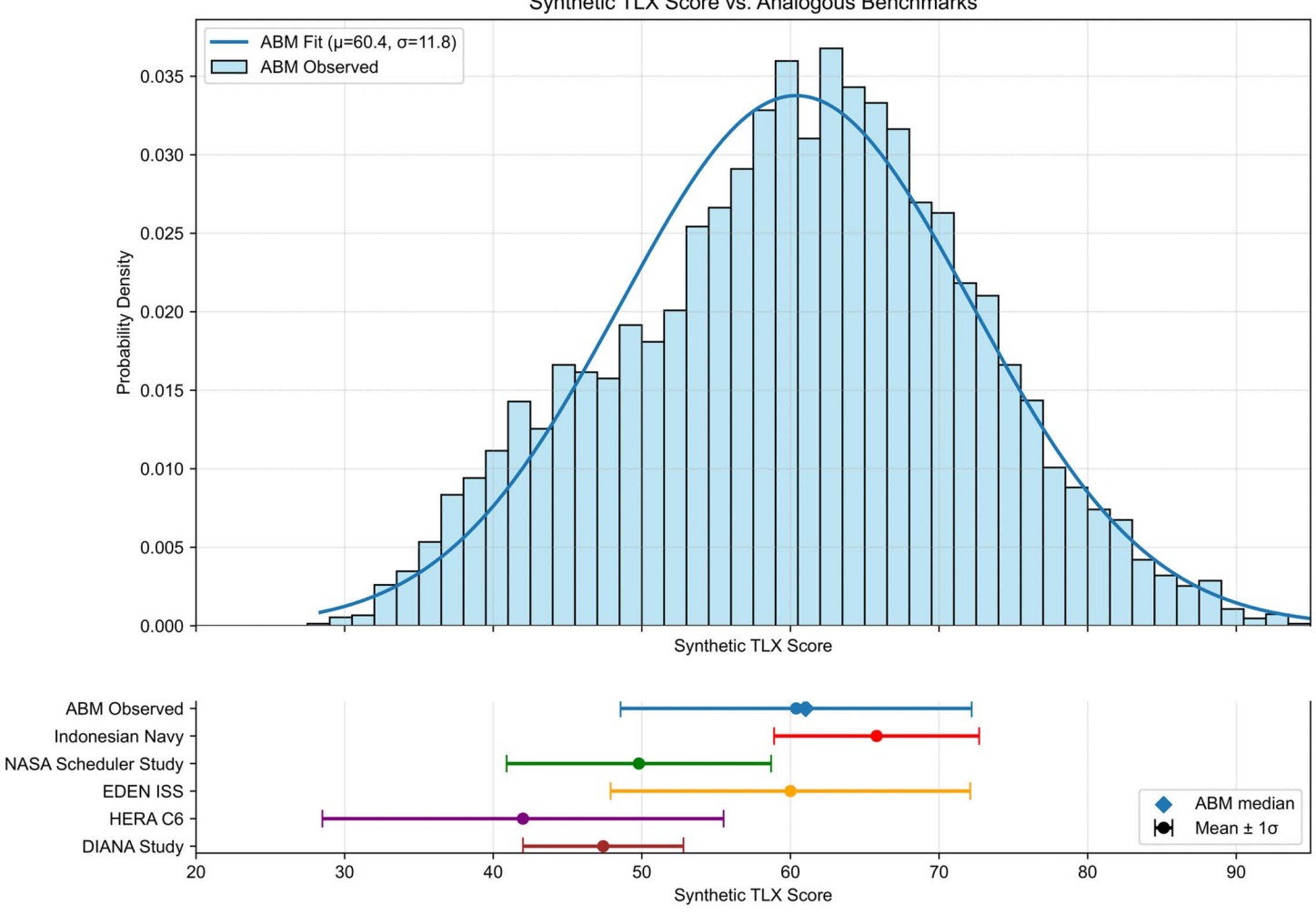

**Fig 7. Synthetic TLX score distribution (initial case).** ABM synthetic TLX scores are shown alongside analogous benchmark distributions, including NASA's HERA, EDEN ISS, and other studies.

**Table 13. Lunar Base ABM synthetic TLX score comparison data.**

| Study Benchmark | Mean | Standard Deviation | Minimum | Maximum | Median |
|---|---|---|---|---|---|
| Lunar Base ABM | 60.4 | 11.8 | 28.4 | 100.0 | 61.0 |
| Indonesian Navy | 65.8 | 6.9 | 58.8 | 74.3 | 64.3 |
| NASA Scheduler | 49.8 | 8.9 | 32.2 | 59.0 | 50.0 |
| EDEN ISS | 60.0 | 12.1 | 36.0 | 77.3 | 60.0 |
| HERA C6 | 42.0 | 13.5 | 20.0 | 60.0 | 45.5 |
| DIANA | 47.4 | 5.4 | 40.2 | 57.5 | 46.8 |

## Scenario analysis

The final section of this paper will demonstrate how to utilize the Lunar Base ABM to support planning and risk assessment for future Artemis missions. The scenarios under consideration involve changes to input parameters relating to

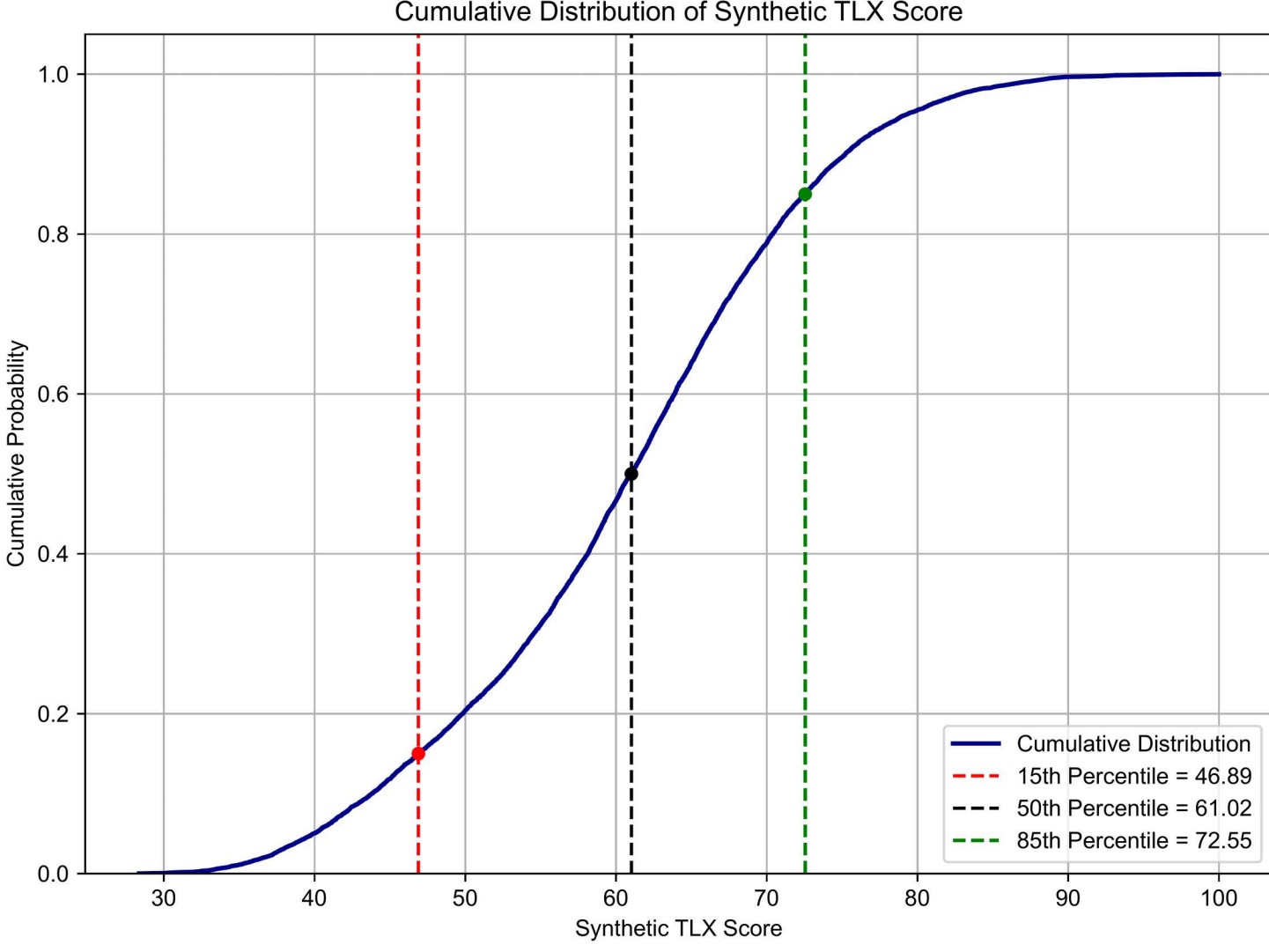

**Fig 8. Synthetic TLX score cumulative distribution (initial case).** Cumulative probability distribution of synthetic TLX scores in the Lunar Base ABM. The vertical lines represent the 15th percentile (46.89), 50th percentile (61.02), and 85th percentile (72.55), which correspond to performance and workload confidence levels.

team composition, exogenous variables, and mission requirements. The first two cases illustrate the impact of changes in the astronaut population, resulting in different teams. Case 1 increases the number of astronauts to a total of 10, evenly divided between the Moon Base and the Gateway. Case 2 enables the transfer of astronauts between the Moon Base and Gateway every 2 weeks, resulting in dynamic team compositions. Cases 3 and 4 involve changes in exogenous parameters, including an increase in unexpected task probability (from Uniform[0.0, 1.0] to 0.90) and learning rate (from 0.85 to 0.95). Case 5 demonstrates the change in mission requirements by extending the total duration from 3 to 6 months, Case 6 explores what happens when the crew experiences the death of an Agent_Astronaut towards the last third of the mission, and Case 7 shows the situation with zero astronaut replacements in the Moon Base or Gateway. Lastly, Cases 8 and 9 represent the minimum and maximum TLX values, respectively, that could be achieved using sensitivity analysis of specific input parameters under the 3 month baseline duration. The best case scenario (i.e., minimum TLX) consists of 6

astronauts, 2 week resupply window, minimal adverse environmental probabilities, and 0.75 learning rate. In contrast, the worst case scenario (i.e., maximum TLX) consists of 4 astronauts, 1 month resupply window, moderate to high adverse environmental probabilities, and 0.95 learning rate. Table 14 summarizes the key differences between the initial base scenario and the nine additional cases used for scenario analysis.

Tables 15-18 show the mean (with percentage change from intial case), standard deviation (Std Dev), minimum (Min), maximum (Max), and median values (Med) for the initial case and scenario cases (i.e., 1–9) of the total task completion, average coping capacity, average tension level, and synthetic TLX score model output.

## Total task completion results

Case 5 has the highest improvement in total tasks completed (+100% on average) because of the increase in mission time to complete activities. For cases with no change in mission requirements, Case 8 had the most positive impact, resulting in a 27% increase followed by Case 1 at +16%, which involves a higher astronaut population. Overall, additional astronauts include more opportunities for specialization that lead to an increase in the optimization of professional skill levels. Case 4 resulted in the most negative impact on the total task completed (−14% on average). A higher learning curve rate means that less efficiency (i.e., learning) is gained with each additional task completed.

**Table 14. Lunar Base ABM initial and scenario cases.**

| Scenario | Description |
| --- | --- |
| Initial | 4 astronauts, 2 rovers, 3-month mission, 0.85 learning rate, baseline values |
| Case 1 | 10 astronauts, evenly split between Moon Base and Gateway |
| Case 2 | Agent_Astronaut transfer every 2 weeks |
| Case 3 | Unexpected task probability increased from Uniform[0.0, 1.0] to 0.90 |
| Case 4 | Learning rate increased from 0.85 to 0.95 |
| Case 5 | Mission duration extended from 3 months to 6 months |
| Case 6 | Agent_Astronaut death |
| Case 7 | Zero Moon Base or Gateway astronaut replacements |
| Case 8 | Minimum TLX parameters from sensitivity analysis |
| Case 9 | Maximum TLX parameters from sensitivity analysis |

**Table 15. Total task completion statistics.**

| Case | Mean | Std Dev | Min | Max | Median |
| --- | --- | --- | --- | --- | --- |
| Initial | 748 | 73 | 451 | 964 | 754 |
| 1 | 867 (+16%) | 38 | 659 | 973 | 871 |
| 2 | 756 (+1%) | 65 | 474 | 930 | 761 |
| 3 | 757 (+1%) | 68 | 434 | 946 | 762 |
| 4 | 644 (−14%) | 65 | 411 | 892 | 645 |
| 5 | 1498 (+100%) | 109 | 1008 | 1830 | 1505 |
| 6 | 716 (−4%) | 70 | 434 | 905 | 721 |
| 7 | 738 (−1%) | 83 | 406 | 943 | 745 |
| 8 | 950 (+27%) | 45 | 714 | 1094 | 954 |
| 9 | 655 (−12%) | 72 | 353 | 921 | 655 |

**Table 16. Average coping capacity statistics.**

| Case | Mean | Std Dev | Min | Max | Median |
|---|---|---|---|---|---|
| Initial | 0.67 | 0.11 | 0.30 | 0.99 | 0.68 |
| 1 | 0.77 (+14%) | 0.05 | 0.57 | 0.94 | 0.77 |
| 2 | 0.72 (+7%) | 0.11 | 0.30 | 0.99 | 0.73 |
| 3 | 0.67 (0%) | 0.10 | 0.29 | 0.97 | 0.68 |
| 4 | 0.69 (+2%) | 0.12 | 0.29 | 0.99 | 0.68 |
| 5 | 0.49 (−27%) | 0.06 | 0.24 | 0.66 | 0.50 |
| 6 | 0.61 (−10%) | 0.13 | 0.33 | 1.00 | 0.61 |
| 7 | 0.66 (−2%) | 0.12 | 0.28 | 0.99 | 0.67 |
| 8 | 0.76 (+12%) | 0.07 | 0.50 | 0.98 | 0.75 |
| 9 | 0.73 (+9%) | 0.11 | 0.35 | 1.00 | 0.71 |

**Table 17. Average tension statistics.**

| Case | Mean | Std Dev | Min | Max | Median |
|---|---|---|---|---|---|
| Initial | 0.37 | 0.20 | 0.00 | 1.00 | 0.37 |
| 1 | 0.16 (−56%) | 0.08 | 0.00 | 0.50 | 0.16 |
| 2 | 0.27 (−27%) | 0.17 | 0.00 | 0.96 | 0.26 |
| 3 | 0.27 (−27%) | 0.17 | 0.00 | 0.99 | 0.25 |
| 4 | 0.35 (−5%) | 0.18 | 0.00 | 0.96 | 0.35 |
| 5 | 0.38 (+2%) | 0.20 | 0.00 | 0.98 | 0.37 |
| 6 | 0.20 (−45%) | 0.17 | 0.00 | 1.00 | 0.20 |
| 7 | 0.51 (+39%) | 0.24 | 0.00 | 1.00 | 0.53 |
| 8 | 0.23 (−38%) | 0.12 | 0.00 | 0.80 | 0.23 |
| 9 | 0.42 (+14%) | 0.21 | 0.00 | 1.00 | 0.44 |

## Coping capacity results

Case 1 has the highest improvement in coping capacity (+14% on average) followed closely by the minimum TLX scenario or Case 8 (+12%) because the astronauts are more successful in completing activities due to specialization and optimization. Cases 5 and 6 have the most negative impact on coping capacity (−27% and −10%, respectively). Coping capacity will decrease over time as humans are separated from their natural habitat (i.e., Earth), including interactions with their family and friends. Moreover, the loss of a close team member during a high-risk space mission can have a substantial adverse effect on crew psychological health.

## Tension level results

Case 1 has the most significant improvement in tension level (−56% on average) followed closely by the minimum TLX scenario or Case 8 (−38%) because more astronauts increase the probability of having compatible or effective personality types working together to reduce interpersonal stress. In contrast, the absence of astronaut replacement during the mission duration (Case 7) results in the highest tension increase of +39% since the team composition remains static and tension will continue to increase over time due to personality incompatibilities and task failures. The maximum TLX scenario (Case 9) also performs poorly with respect to tension levels resulting in an increase of +14%.

**Table 18. Synthetic TLX score statistics.**

| Case | Mean | Std Dev | Min | Max | Median |
|------|------|---------|-----|-----|--------|
| Initial | 60.38 | 11.82 | 28.36 | 100.00 | 61.02 |
| 1 | 41.77 (−31%) | 4.57 | 28.26 | 63.00 | 41.77 |
| 2 | 54.51 (−10%) | 10.44 | 28.36 | 95.37 | 54.02 |
| 3 | 56.31 (−7%) | 9.83 | 30.54 | 95.14 | 55.70 |
| 4 | 65.93 (+9%) | 10.79 | 31.40 | 100.00 | 66.77 |
| 5 | 57.20 (−5%) | 8.00 | 37.44 | 83.58 | 57.06 |
| 6 | 58.79 (−3%) | 10.85 | 30.41 | 100.00 | 58.91 |
| 7 | 66.91 (+11%) | 13.64 | 26.54 | 100.00 | 67.89 |
| 8 | 42.44 (−30%) | 6.03 | 26.18 | 71.09 | 42.89 |
| 9 | 66.26 (+10%) | 12.55 | 28.00 | 100.00 | 68.06 |

### Synthetic TLX score results

Case 1 has the most significant improvement in synthetic TLX score (−31% on average) followed closely by the minimum TLX scenario or Case 8 (−30%) because more astronauts increase the probability of personality type compatibility and optimize for professional skill levels that will lead to less psychological strain and more successful task completion. Alternatively, increasing the learning curve rate (Case 4), prohibiting astronaut replacements (Case 7), and the maximum TLX scenario (Case 9) will cause additional psychological stressors and task inefficiencies on the astronauts, resulting in a synthetic TLX score increase of 9%, 11%, and 10%, respectively.

### Discussion

From the Monte Carlo simulations, the Lunar Base ABM produced consistent and emergent distributions in task productivity, coping capacity, tension level, and synthetic TLX scores (Figs 4-7). The simulated astronaut crew completed approximately 20% of baselined tasks (i.e., not including unexpected activity) with coping capacity and tension output values aligning closely to the analogous Antarctic studies. Scenario analysis shows that increasing crew size results in optimizing skill specialization and increasing the chance of teamwork personality compatibility. In contrast, prolonged mission durations, higher learning rates, and the absence of astronaut replacements introduces additional psychological stress resulting in a decrease of task performance and increase of overall TLX scores. Examples of actionable insights that can be obtained from the model include optimizing crew size, resupply frequency, and mission duration to achieve the acceptable TLX score and productivity rate. Investment in technical skills training can also be justified to offset the steep learning curve slopes. These results demonstrate how the integration of cognitive, emotional, social, and environmental factors in the Lunar Base ABM can be used to study team dynamics and improve future lunar mission planning.

There are certain model limitations to acknowledge that can be improved in future iterations of the Lunar Base ABM. First, the model does not include physiological effects that astronauts experience during typical space missions. As the mission duration increases pass the three month baseline, the physical conditions of the human body become more stressed due to the influence of extreme environmental and psychological factors. Second, the model does not incorporate communication delays between the Earth, Moon Base, and Gateway that is known to increase operational stresses and can have a negative impact to task performance and emotional health. Lastly, the model can benefit significantly with more data relating to actual historical TLX scores, coping capacity, interpersonal relationships, teamwork dynamics, and the lunar and analogous environment, which can be used for calibration of input parameters and output validation.

Future ideas to explore and enrich this ABM can include: refining the integrated disposition Eq 3 to account for 1) the impact on coping capacity and tension during interactions outside of work-related tasks, 2) during physical location changes (e.g., astronaut transfer between the Moon Base and the Gateway), or 3) in the exploration and interactions

with the extreme environments in space. The space mission can also be expanded to include multiple lunar habitats and a complex logistic infrastructure on the Moon and in Moon's orbit. Additionally, the model can be expanded to analyze energy and resource production and consumption, while assessing equipment failure dynamics to promote sustainability. Lastly, the ABM can be used as a building block for a developing a space economy consisting of mining operations, lunar industrial base/supply chain, manufacturing, and a competitive market for in-situ resource utilization (ISRU), rare earth elements (REEs), rocket fuel, and new technologies (e.g., ZBLAN optical fibers).

Our model is not only a benchmark for human space missions (currently focused on the Moon), but represents also an interesting case of emergence in highly controlled and designed top-down phenomena, that warrants further methodological exploration.

## Conclusion

NASA is planning to conduct several crewed missions to the Moon (i.e., Artemis III, IV and V) that will send the first humans to explore the region near the Lunar South Pole and put the Gateway space station in lunar orbit, in the late 2020s or early 2030s, which are necessary stages to establish a human presence on the Moon. This opens up a new area of research opportunities for social, economic, and, in general, human factors, in the lunar environment, and in space, in general. By including a complex systems approach to explore the emergence of social phenomena in space, and using ABMs as a methodology for both top-down and bottom-up simulations, we can answer very important questions about the role of human factors for space missions success. The Lunar Base ABM is the first step to understanding how astronauts with different skill levels and personality types can work together to operate and sustain a fully functional lunar habitat. Through this framework, the Lunar Base ABM can provide a powerful tool for mission planners to test scenarios under different team configurations, ensuring that future Artemis missions are safer, more efficient, and better equipped to handle the unknowns of lunar exploration.

## Author contributions

**Supervision:** Anamaria Berea.

**Writing – original draft:** Raymond Vera, Anamaria Berea, William G. Kennedy.

**Writing – review & editing:** Raymond Vera, Anamaria Berea, William G. Kennedy.

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
