## [Decision Letter · Decision Letter 0]

31 Oct 2025

PONE-D-25-43108

Lunar base agent-based modeling - A benchmark for simulating crewed space missions

PLOS ONE

Dear Dr. Vera,

Thank you for submitting your manuscript to PLOS ONE. After careful consideration, we feel that it has merit but does not fully meet PLOS ONE’s publication criteria as it currently stands. Therefore, we invite you to submit a revised version of the manuscript that addresses the points raised during the review process.

We look forward to receiving your revised manuscript.

Kind regards,

Birendra Mishra, DVM, PhD

Academic Editor

PLOS ONE

Journal Requirements:

“GMU ORIEI Award no. 102264”

5. Please expand the acronym “GMU ORIEI” (as indicated in your financial disclosure) so that it states the name of your funders in full.

6. Please provide a complete Data Availability Statement in the submission form, ensuring you include all necessary access information or a reason for why you are unable to make your data freely accessible. If your research concerns only data provided within your submission, please write "All data are in the manuscript and/or supporting information files" as your Data Availability Statement.

Additional Editor Comments:

After a thorough evaluation, we recognize its value; however, this manuscript requires significant revisions for further consideration. Please enhance the figures' resolution and font size to improve readability, as these figures currently do not meet publication standards.

Reviewers' comments:

Reviewer's Responses to Questions

**Comments to the Author**

1. Is the manuscript technically sound, and do the data support the conclusions?

Reviewer #1: Yes

Reviewer #2: Partly

Reviewer #3: Partly

Reviewer #4: Yes

Reviewer #5: Yes

Reviewer #6: Partly

2. Has the statistical analysis been performed appropriately and rigorously?

Reviewer #1: Yes

Reviewer #2: No

Reviewer #3: I Don't Know

Reviewer #4: I Don't Know

Reviewer #5: N/A

Reviewer #6: Yes

3. Have the authors made all data underlying the findings in their manuscript fully available?

Reviewer #1: Yes

Reviewer #2: Yes

Reviewer #3: No

Reviewer #4: No

Reviewer #5: Yes

Reviewer #6: Yes

4. Is the manuscript presented in an intelligible fashion and written in standard English?

Reviewer #1: Yes

Reviewer #2: Yes

Reviewer #3: Yes

Reviewer #4: Yes

Reviewer #5: Yes

Reviewer #6: Yes

5. Review Comments to the Author

Reviewer #1: The paper describes the agent-based model enabling simulation of long-term effects of human factor and interactions and their impact on various mission success parameters, such as task performance and completion, coping capacity of the crew or their coping capacity.

The topic is interesting from the point of view of the mission planning and operations, potentially lowering the risk associated with with unexpected events occurence or poor characterological crew composition. However, to increase the model reliability, it is required to broaden the dataset used to verify the model parameters. Considering the above, I recommend minor revisions of the submitted article, with a focus on highlighting model's limitations and unceirtanities and presenting further development directions in the text body.

General remarks:

I think the model would benefit greatly from considering the data from analogous habitat studies in underwater simulants, cave studies as well as from isolation studies, i.a. performer during the pandemic. Their consideration might help minimise the effects of limited dataset, providing insights not gathered elsewhere. Is there any particular reason you decided not to use such data? (10.1016/j.actaastro.2024.10.044, 10.1016/j.actaastro.2021.04.003, 10.1027/1016-9040/a000453)

Moon communication delay will be an additional stress factor for the crew, additionally increasing the psychological pressure in high-risk situations, when fast decisions are required (~2,5 sec could be even a life-death matter).

Space missions are putting the crew in greater danger than any ground-based habitat: the lack of oxygen in space scenario makes life support system failure even more critical (except for the underwater habitats). Did you consider including the additional "danger factor" for the dataset, so the model would consider the ground-based results as the lower limit of the stress factor associated with the environment?

You mention the EDEN ISS in your paper, did you consider including the positive effects of in-situ plant production on psychological condition of the crew? (10.1007/978-3-030-20503-4_71)

Regarding the areas for your further consideration (not necessarily to be included in the reviewed paper, but for the future model development):

The interlink between the psychological and physiological health should not be overlooked by the model.

Loss of the crew member is additional factor to impact the psychological health - depending on the reason lowering (well-suited in terms of DISC personalities) or improving (poorly-suited in terms of DISC personalities) during standard crew member change. A separate scenario could consider fatal loss of crew member.

In Author summary section, when mentioning the Agent_Zero framework for the first time, the citation is missing.

L2: The first American orbital space flight took place in 1962 (Mercury-Atlas 6 mission), with astronaut John Glenn onboard. Manned space program in America started in 1961 with 2 suborbital flights within the Mercury program, following the Soviet's Wostok program. In 1986 the first manned flight beyond the Earth's orbit, heading to the Moon, took place during the Apollo 8 mission.

L7-9: Initially JAXA and CSA were not involved in station building. Maybe rephrasing the sentence would be better, to show both that all the 5 partners together develop ISS today, but the ISS' history started in 1990s.

L80: i in Agent i should be cursive probably

L87, 88: citations to the Latan´e-Darley experiment + social media impact on 2011 Arab Spring uprising needed.

L95: "hand-picked by NASA" - I would prefer the phrase "by space agencies", not pointing NASA exclusively. As the Lunar base will finally be used by astronauts of various nationalities (e.g. from European countries), it's worthy to highlight that all the space agencies put equal efforts to prepare their crews for extreme conditions of space missions.

L192-194: source of information on Perserverance power source and its lifetime needed

L202: Gateway will not be on the Moon but on Moon's orbit, please try to rephrase the sentence ("(...) a theoretical habitat environment on the Moon (...)")

Table 2, physical-health attribute: does the value degradate over time? The impact of lowered/microgravity on muscle mass, bone density etc. on humans should affect the parameter additionally over time.

personality_type attribute: what about all the in-between DISC personality types?

Table 4, unexpected_task_prob attribute: where from comes the starting value? Could you provide some reference here?

Table 6, shipment of resources: Does the probability of shuttle accident reflects the probability of not getting the resupply and is included the model?

Table 9: Is the number of hours of sleep fixed permanently or can it be shortened by e.g. unexpected accident? Can the simulation include the neccesity to work past normal time and how it affects Astronaut's health? (e.g. by slowering the recovery)

L333-338: Are the mentioned changes reflected by changes in Crew stress levels?

L451: The model is generally well-described, but I encourage you to share the code publicly (e.g. via GitHub), to increase its visibility and chances for reuse/results reproduction: 10.1016/j.envsoft.2020.104873

L599-607: This limitation should be expressed more clearly. For now, it is the weakest point of the model as tension level is stictly related to interpersonal dynamics which the model tries to explore and define its dependencies with other stressors. Establishing the rationale of such model's behaviour should be described in more detail, maybe cosider even creating a separate chapter where all the relevant aspects could be discussed.

L644: CL - I assume the abbreviation stands for Confidence Levels? Please add full name with the first abbreviation appearance

Discussion section: Please consider to broaden the section, especially to point out the potential data sources for the model improvement (so not pnly the analogue habotats and Antarctica studies but also all the rest mentioned above). Put it in the context of how such data translates to space conditions and what kind of studies would be needed to improve the model.

Conclusions section: Include the current model limitations, especially the interpersonal tension level.

References: Reference list in over 1/3 are websites - please add the missing citation where applicable and maybe try to provide additional references wherever possible for the Internet ones?

Figure 1, Incidental variables: typo, should probably be environmental changes (a is missing).

Reviewer #2: While I believe there is significant merit and utility In this modeling system, the paper requires an overhaul, or reframing regarding it’s context and goals. I recommend this paper for acceptance pending major revision. However, I would like it to be clear that I believe the contents of the paper are strong, the primary revisions are reshaping the expectations of the readership. My comments are below:

I suggest refining the purpose of the paper. It reads heavily like a white paper explaining the methods to design the ABM, but also discusses some basic results of the model. I suggest making it clearer to readers that this is a methodology paper and provide proof of validity of the model and publish the results separately, or significantly reducing the methodology of how the model was built, and focus more on the output/results of the model. This is a strong paper regardless of the final direction.

Methods and materials:

Authors define the Lunar Base as a complex system twice, this is not necessary.

The models are used for both human and system-wide inputs. The setup of this section is somewhat confusing. I suggest restructuring, and using less parenthetical commentary, such as, “…as a matter of fact.”

Depending on the audience you want to attract, it may be worthwhile to put the application of the Agent_Zero model and the fight or flight response into a diagram to allow those who are unfamiliar to get a better understanding.

Also consider further describing the 2011 Arab Spring uprising- as readers may not be familiar with it or ‘affective homophily.’

Consider moving the limitations of Epstein’s model to the conclusions and limitations.

Define the Rescorla-Wagner rule.

Figures 2 & 3 are too blurry to read or interpret. All the graphs and figures are blurry, but those two are unreadable.

Table 9- please confirm the schedule, is this theoretical, or based on the literature? I do not think that the time dedicated to exercise has been officially determined.

Results and discussion:

Remove figures 4-6 and the discussion around them regarding the normal distributions to visualize the data, or run the analyses again with the data and their true distributions. It is not reasonable to display these and then utilize them as results. This reads as poor scientific methodology, whereas authors clearly put immense intention into building the model.

The discussion is far too short. Please expand. Put limitations of the model in this section instead of the methods. If keeping results in the paper, add how this information can be applied to future missions, etc.

Reviewer #3: The authors conducted simulated lunar space missions in an agent-based model (ABS). The ABS explored factors and simulated crew interactions that influenced the success of Artemis like scenarios. The objective as stated by the authors was to simulate a theoretical habitat environment on the Moon for astronauts to perform relevant space mission tasks, Success was measured by task performance.

The overall concept and approach of this study is commendable and the results of the model simulations are very interesting. However, the organization of the manuscript can use improvement to facilitate overall readability. Also, the presentation of the methods and results are somewhat limited and the discussion only touches the surface of how the data could be used during mission planning.

It would greatly benefit the manuscript if the authors revisited the order in which material is presented. On first read, the first half of the manuscript is very difficult to follow and does not provide the context needed to understand what the authors set out to investigate. On page 3, it is still not clear whether the ABM would be predominantly focused on factors related to the ‘lunar environment’ or ‘human factors' (or both, or something else). It is not until around the discussion of the design concept (page 14) that the context is made clear.

Here are two specific examples where context would help, but the authors should go through the entire paper for other cases as well.

Page 2 “The lunar environment also encompasses endogenous and exogenous factors… …reinforced by positive and negative feedback loops…” This finally made more sense on page 13 after the reference to Figure 3.

Page 3 “…heterogenous actors and the flow of information, energy, and resources.”

The manuscript leading up to the design concept section is somewhat vague and perhaps assumes the reader already understands the approach and components of the ABM. For instance, the attributes of the agents are described between pages 3 and 6, before their importance to the ABM is understood. The ‘agents’ are finally introduced in the middle of a paragraph on page 7. After reading the design concept section, it was much easier to go back and understand the pages leading up to it.

The use of terms with specific vs. generic meanings should be reduced throughout the manuscript as much as possible unless the context can be made clear. i.e. terms such as “skills”, “attributes”, “tasks”, ‘activities’, ‘endogenous’, ‘exogenous’ ‘parameters’, ‘variables’, ‘components’, ‘factors’, ‘interactions’. Also check for consistency, for example: ‘Moon Base’ vs ‘Lunar Base’

Tables 2 through 8. These tables are very helpful in providing context. The column labeled ‘Starting Value (or range)’ needs some clarification. The ranges are more helpful than only providing a starting value. It is understood (or assumed), for instance, that “base_waste” in Table 4 should only be positive values, but is there a max value associated with this attribute or does a missing range mean there is no upper limit? Also in Table 4: Starting location is set to ‘Moon’. What is the reason for including this attribute; are other locations possible in this case?

Page 12: “Together, these four components…” Not clear to which four components it refers.

The fonts in Figures 2 and 3 are too small to read in standard letter format.

The beginning of the results section seems to be more of a method description. More detail should be provided about the different cases that were. Table 11 provides some info, but it is insufficient to discern what parameters are the same vs. different between each of the cases. For instance, how many astronauts in cases 2-5. What are the locations in cases 2-5. This table could be replaced with a table that provides the relevant parameters consistently for all cases.

The final subsections within the results may be more suitable for the discussion. These interpretations could be further expanded. For instance, based on this model, do the authors conclude anything regarding optimum crew sizes, transfer rates, mission duration, tension reduction, etc? It may be premature to extrapolate beyond the limits of this model, but it could still make for an interesting discussion.

In summary, the content of the manuscript is very interesting, but the organization of the paper should be improved. Furthermore, the methods and results of the cases that were explored should be outlined in more detail to make case comparisons easier and facilitate the interpretation of the findings.

Reviewer #4: Comments:

Where is the data available?

page 2, line 25: "While engineering and technology innovation drives space missions,..." - This affirmation is subjective. What drives space exploration can be many things. Please repharase or remove. The rest of the phrase is fine. I suggest: "While engineering and technology innovation is necessary for space missions, understanding

human and operational dynamics is crucial for mission success. Even more so if the objective is to establish long term presence on the Moon."

Page 2, line 34, The authors mention NASA's human factors and behavioral research. Is there a reference(s) to this research? I would like to know which specific research is this model based on. This is also necessary for verification and reproduction of the method.

Page 3, line 49. This phrase has several explanations inside parenthesis () which are expendable because it do not add essential information and add reading complexity. I suggest to remove them along with their content. Besides, the phrase that follows explains what are the bottom-up and -down layers.

I have not found further remarks on the article. I would like to see how this model fares when compared to new ground simulations. Looking forward for your future work.

Reviewer #5: 1) Design of the study and the introduction was planned properly and summarized explicitly.

2) Using and integrating computational approach into this sort of study is a plus, to further evaluate ABMs, in my opinion.

3) This effort also takes advantage from its multianalytical approach and contributes to literature positively.

4) It has to be accepted that this type of work has many different parameters (participants’ physiological, cognitive, psychological, social and similar etc.) that should be considered but difficult to integrate with small number of participants. Thus, maybe a larger dataset should be used to evaluate such a study here, but this is very rare to find that number of participants due to ethical situations and volunteer groups. Even if the authors do not have such option, this is understandable, of course.

5) I strongly suggest authors integrate much more explanatory diagrams/charts for what they are presenting. Rather than boxes and interactions/connections

6) Data presented in Figs. 2 and 3 are not readable.

7) Conclusion is short and understandable. However, it would be much better to enhance this section a bit more to give the idea and the importance of the effort spent.

8) If possible, readers would want to see the Python codes to reproduce the results and/or develop the code.

Reviewer #6: Thank you for the opportunity to review your paper entitled “Lunar Base Agent-Based Modeling – A Benchmark for Simulating Crewed Space Missions.” This manuscript presents a sophisticated and well-structured agent-based model (ABM) designed to simulate the complex interactions between human, environmental, and operational factors in lunar mission scenarios. I found the study to be thoughtful, well-motivated, and clearly written, with a strong methodological foundation and meaningful implications for future Artemis and lunar base planning. The integration of human behavioral modeling,particularly coping capacity, tension, and workload dynamics represents an important contribution to the growing field of space systems simulation.

1. The paper presents a strong and well-structured ABM, but the introduction could better emphasize what makes this model unique. A short paragraph contrasting it with previous Mars or lunar simulations and explaining how it integrates psychological variables such as coping and tension would make its novelty clearer.

2. The adaptation of Epstein’s Agent Zero is a creative foundation, though the mechanics need clarification. It would help to specify how the cognitive, affective, and social parameters were scaled, how often they update, and what range of thresholds (τ) was tested. Since the logic is multiplicative rather than additive, a short equation or pseudo-code would make the implementation easier to follow.

3. The description of task thresholds and teamwork needs more detail. Readers should know how the model aggregates multi-skill cooperation,whether individual skills are added, averaged, or multiplied, and what specific threshold values were used for task success.

4. The DISC-based personality reactivity coefficients (D = 0.3, C = 0.4, S = 0.5, I = 0.7) seem reasonable but appear judgment-based. A brief justification or sensitivity check would confirm robustness. Clarify whether these penalties occur once per event or accumulate over time.

5. The learning-curve formulation is well motivated but incomplete. Please indicate which slope values were actually applied to each task category, whether learning occurs at the individual or team level, and whether it affects time-to-complete or probability of success.

6. The NASA TLX integration is one of the paper’s strengths but remains under-explained. It would help to map TLX subscales (mental, physical, temporal, effort, frustration, performance) to model variables such as task complexity or coping capacity and to specify if TLX is updated per task or per day and how subscales are weighted. Explicitly linking Figure 1 to these components would turn it into a real part of the model rather than a reference diagram.

7. Stochastic parameters should include their full distributions and time bases. The reported moonquake probability (0.00096 day⁻¹) seems low and might need a magnitude qualifier, while the shuttle accident rate (0.06456) should be clarified as per-flight, not per-day. These details are essential for reproducibility.

8. The model’s handling of emergency events could be clearer. A brief explanation of whether astronauts abort ongoing EVAs or finish the current hour before reacting would make the resilience logic more transparent.

9. Resource and energy relationships deserve one line of clarification, whether energy limits task execution or if resource production draws power from the system. If not yet modeled, noting that it’s planned would prevent confusion.

10. Because the study aims to provide a benchmark, the authors should consider including the code or at least a full parameter appendix listing every stochastic input, range, and time unit so others can replicate the results.

11. In the results, the baseline case reports a 20 % task completion rate, but breaking down critical versus non-critical tasks would show whether mission performance risks are hidden within aggregated totals.

12. The histograms fitted with normal curves are useful visuals, but since ABM outputs are rarely normal, it would be better to report medians and interquartile ranges or note that the normal fits are illustrative only.

13. Coping aligns well with Antarctic analogs, but tension is unrealistically low. Acknowledge that the baseline excludes strong social stressors (no transfers, stable schedules) and that conservative reactivity settings likely suppress tension variability.

14. The TLX calibration using Equation 4 is clear but should restate that it only rescales outputs and does not alter simulation dynamics. Clarify what the threshold value (≈67.7) actually represents, an empirical percentile or an acceptable workload limit.

15. In the scenario analysis, a brief mechanism summary would sharpen interpretation: Case 1’s larger crew improves specialization and compatibility; Case 2’s frequent transfers disrupt cohesion; Case 3’s extra tasks raise stress; Case 4’s higher learning slope weakens efficiency; Case 5’s extended mission increases fatigue. Showing monthly trends for Case 5 would reveal how stress accumulates.

16. Along with percentage changes, include absolute deltas and standardized TLX shifts to make comparisons across scenarios clearer.

17. The discussion would benefit from connecting findings to actionable insights,crew sizing as an efficiency lever, transfer cadence as a stability factor, and training investment to offset steep learning slopes, and from acknowledging the optimistic baseline assumptions that reduce tension and risk.

Overall, the paper is technically impressive, I really enjoyed I, I think strengthening these parts I mentioned above, particularly clarifying implementation details, TLX mapping, and scenario mechanisms will make the paper both more transparent and more directly useful for future mission planning and modeling studies.

6. PLOS authors have the option to publish the peer review history of their article (what does this mean?). If published, this will include your full peer review and any attached files.

Reviewer #1: No

Reviewer #2: No

Reviewer #3: No

Reviewer #4: **Yes:** Dr. Pedro M. R. Reis

Reviewer #5: No

Reviewer #6: No

---

## [Author Response · Author response to Decision Letter 1]

16 Dec 2025

All responses to reviewer and editor comments are included in the submitted document, "'Response to Reviewers."

---

## [Decision Letter · Decision Letter 1]

23 Apr 2026

Lunar base agent-based modeling - A benchmark for simulating crewed space missions

PONE-D-25-43108R1

Dear Dr. Vera,

We’re pleased to inform you that your manuscript has been judged scientifically suitable for publication and will be formally accepted for publication once it meets all outstanding technical requirements.

Kind regards,

Babak Aslani

Academic Editor

PLOS One

Additional Editor Comments (optional):

Reviewers' comments:

Reviewer's Responses to Questions

**Comments to the Author**

1. If the authors have adequately addressed your comments raised in a previous round of review and you feel that this manuscript is now acceptable for publication, you may indicate that here to bypass the “Comments to the Author” section, enter your conflict of interest statement in the “Confidential to Editor” section, and submit your "Accept" recommendation.

Reviewer #1: All comments have been addressed

Reviewer #4: All comments have been addressed

Reviewer #5: All comments have been addressed

2. Is the manuscript technically sound, and do the data support the conclusions?

Reviewer #1: Yes

Reviewer #4: Yes

Reviewer #5: Yes

3. Has the statistical analysis been performed appropriately and rigorously?

Reviewer #1: Yes

Reviewer #4: I Don't Know

Reviewer #5: Yes

4. Have the authors made all data underlying the findings in their manuscript fully available?

Reviewer #1: Yes

Reviewer #4: Yes

Reviewer #5: Yes

5. Is the manuscript presented in an intelligible fashion and written in standard English?

Reviewer #1: Yes

Reviewer #4: Yes

Reviewer #5: Yes

6. Review Comments to the Author

Reviewer #1: The revision of the paper adresses all the points I find relevant, the source code was made publicly available and the discussion section includes model's limitations. Thank you!

Reviewer #4: (No Response)

Reviewer #5: all my previous queries were addressed properly. Figures 2 and 3 are much more readable in good resolution. Conclusion is still short but OK.

7. PLOS authors have the option to publish the peer review history of their article (what does this mean?). If published, this will include your full peer review and any attached files.

Reviewer #1: No

Reviewer #4: **Yes:** Dr. Pedro M. R. Reis

Reviewer #5: **Yes:** Ozan Unsalan

---

## [Editor Report · Acceptance letter]

PONE-D-25-43108R1

PLOS One

Dear Dr. Vera,

I'm pleased to inform you that your manuscript has been deemed suitable for publication in PLOS One. Congratulations! Your manuscript is now being handed over to our production team.

Kind regards,

on behalf of

Dr. Babak Aslani

Academic Editor

PLOS One